



# A tracer release experiment to investigate uncertainties in drone-based emission quantification for methane point sources

Randulph Morales[1,4], Jonas Ravelid[1], Katarina Vinkovic[2], Piotr Korbeń[3], Béla Tuzson[1], Lukas Emmenegger[1], Huilin Chen[2], Martina Schmidt[3], Sebastian Humbel[1], and Dominik Brunner[1]

[1]Laboratory for Air Pollution / Environmental Technology, Swiss Federal Institute for Materials Science and Technology, Empa, Dübendorf, Switzerland
[2]Centre for Isotope Research, University of Groningen, Groningen, Netherlands
[3]Institute of Environmental Physics, University of Heidelberg, Heidelberg, Germany
[4]Institute for Atmospheric and Climate Science, ETH Zürich, Switzerland

**Correspondence:** Dominik Brunner (dominik.brunner@empa.ch)

**Abstract.** Mapping trace gas emission plumes using in-situ measurements from unmanned aerial vehicles (UAV) is an emerging and attractive possibility to quantify emissions from localized sources. Here, we present the results of an extensive tracer-release experiment in Dübendorf, Switzerland, which was conducted to develop an optimal quantification method and to determine the related uncertainties under various environmental and sampling conditions. Atmospheric methane mole fractions

were simultaneously measured using a miniaturized fast-response Quantum Cascade Laser Absorption Spectrometer (QCLAS) and an Active AirCore system mounted on a commercial drone. Emission fluxes were estimated using a mass-balance method by flying the drone-based system through a vertical cross-section downwind of the point-source perpendicular to the main wind direction at multiple altitudes. A refined kriging framework, called cluster-based kriging, was developed to spatially map individual methane measurement points into the whole measurement plane, while taking into account the different spatial scales

between background and enhanced methane values in the plume. We found that the new kriging framework resulted in better quantification compared to ordinary kriging. The average bias of the estimated emissions was $-1\%$ and the average residual of individual errors was $54\%$. Direct comparison of QCLAS and AirCore measurements shows that AirCore measurements are smoothened by $20\,\mathrm{s}$ and temporally shifted and stretched by $7\,\mathrm{s}$ and $0.06$ seconds for every second of QCLAS measurement, respectively. Applying these corrections to the AirCore measurements and successively calculating an emission estimate shows

an enhancement of the accuracy by $3\%$ as compared to its uncorrected counterpart. Optimal plume sampling, including the downwind measurement distance, depends on wind- and turbulence conditions and it is furthermore limited by numerous parameters such as the maximum flight time, and the measurement accuracy. Under favorable measurement conditions, emissions could be quantified with an uncertainty of $30\%$. Uncertainties increase when wind speeds are below $2.3\,\mathrm{m\,s^{-1}}$ and directional variability is above $33°$, and when the downwind distance is above $75\,\mathrm{m}$. In addition, the flux estimates were also compared to

estimates from the well-established OTM-33A method involving stationary measurements. A good agreement was found, both approaches being close to the true-release and uncertainties of both methods usually capturing the true-release.



# 1 Introduction

Methane emissions from localized sources such as oil and gas production facilities are often caused by leakage giving rise to highly uncertain emission fluxes with high spatial and temporal variability (Kemp et al., 2016; Fox et al., 2019). A significant disparity was observed, for example, between facility-observed bottom-up emission inventories and a more traditional component-based emission inventory (Brandt et al., 2014; Alvarez et al., 2018). Observation-based estimates from the US indicate that emissions from oil and gas are underestimated in official emission inventories (Gurney et al., 2021). Further measurements of leakage rates from oil- and gas-production facilities in other regions of the world such as those conducted during the ROMEO measurement campaign in Romania (Röckmann and team, 2020), are therefore essential to validate and improve current estimates.

A broad range of methods of methane emission quantification for facility-scale sources has been developed, which includes ground-based thermal imaging (Gålfalk et al., 2016), aircraft remote sensing (Frankenberg et al., 2016; Kuai et al., 2016; Thorpe et al., 2016), chamber sampling (Kang et al., 2014; Yver Kwok et al., 2015), ground-based tracer-release correlation (Lamb et al., 2015, 2016; Omara et al., 2016; Roscioli et al., 2017; Feitz et al., 2018; Fjelsted et al., 2020) and Gaussian plume matching (Ars et al., 2017; Bakkaloglu et al., 2021). Some of these methods, e.g., tracer-release correlation, are quite accurate but expensive, intrusive, and time-consuming, while other methods suffer from large, poorly quantifiable uncertainties.

An emerging and attractive approach to quantify emissions from point sources, or more generally from spatially localized sources, involves deploying integrated unmanned-aerial-vehicle (UAV) systems capable of measuring atmospheric trace gas concentrations. The most common ways of measuring methane from UAVs include: 1) collection of ambient air samples using on-board storage equipment and subsequent analysis of the samples with instrumentation on the ground (Chang et al., 2016; Greatwood et al., 2017; Andersen et al., 2018), 2) live analysis of air samples pumping air into a long tube connected to a ground based analyzer (Brosy et al., 2017; Shah et al., 2019), and 3) in-situ reporting of measurements using an analyzer mounted on the drone (Berman et al., 2012; Nathan et al., 2015; Golston et al., 2017; Martinez et al., 2020; Tuzson et al., 2020). Small UAVs with payloads of a few kilograms are affordable, versatile, and much more easy to deploy compared to larger drones or aircraft. UAVs allow transecting the plume over its entire vertical and horizontal extent, which reduces the dependence on assumptions on horizontal and vertical dispersion compared to ground-based mobile or stationary measurements that only capture a small portion of the plume.

Although UAV-based methane measurements are gaining popularity, systematic studies on testing and comparing different quantification methods and analyzing the different sources of uncertainty are still sparse (Golston et al., 2018; Yang et al., 2018; Shah et al., 2019). The main goal of this study is to develop an improved strategy to quantify local methane sources using UAV measurements, and to test this strategy on UAV measurements obtained downwind from sources with known fluxes. It is crucial to test a new quantification technique with a set of sources with a known release before applying the technique to sources with unknown emissions (Feitz et al., 2018; Shah et al., 2020). To this end, we designed the MethAne Tracer Release EXperiment (MATRIX), where a series of controlled and partly blind methane releases were performed from 09 February to 14 March 2020 in Dübendorf, Switzerland. Methane mole fractions were measured using a drone-based sensor (Tuzson et al., 2020) and





an active AirCore system (Andersen et al., 2018). Adopting the mass-balance approach, the UAV was flown downwind of the source perpendicular to the main wind direction at different vertical levels to derive emission fluxes. In this study, we describe a novel quantification approach and report on its capability to reproduce known emissions. Furthermore, we investigate this approach and its sensitivity to different measurement configurations, and provide recommendations for an optimal sampling.

The new drone-based quantification approach presented here was developed to support the ROmanian MEthane Emissions from Oil and gas (ROMEO) campaign that was taking place in September and October 2019. With $415.60\,\mathrm{ktCH_4}$ per year, Romania has one of the highest per-capita methane emissions from the energy sector in the European Union, according to the latest UNFCCC 2018 Report. This emission estimate was mainly derived using prescribed Tier-1 emission factors following the IPCC guidelines for national reporting, which are both non-country specific and quite uncertain. The ROMEO campaign

was, thus, put into action to investigate the accuracy of this estimate. Eight ground measurement teams, including our drone-system, were deployed to quantify methane emissions from over 1000 oil- and gas-production facilities (Röckmann and team, 2020). Reported emissions from drone-based measurements collected in the western region of Wallachia, Romania during the ROMEO campaign were generated using the quantification approach developed in this study.

        In this paper, we give first an overview of the instruments used in the tracer release experiment (Sect. 2), followed by the

details regarding the setup of the experiments and the mass-balance approach in Sect. 3. The data treatment and interpolation schemes applied to the measurements of both methane and wind are discussed in Sect. 4. Quantification results from the tracer release experiments are presented in Sect. 5.

## 2   Instruments

The in-situ measurements of atmospheric $CH_4$ mole fractions were performed by using two different techniques: i) a lightweight

laser absorption spectrometer and ii) an active AirCore system. These devices were mounted beneath a commercial hexacopter (Matrice 600, DJI), equipped with a RTK-GPS receiver (NEO-M8P-2, SparkFun) for accurate positioning of the drone in all three dimensions. The integrated system, illustrated in Fig. 1, weighs about $13\,\mathrm{kg}$, of which the payload is around $3\,\mathrm{kg}$ and can have a maximum flight time of $20\,\mathrm{min}$.

### 2.1   Quantum Cascade Laser Spectrometer (QCLAS)

The in-situ airborne analyzer, developed at Empa, is a compact and lightweight mid-IR laser absorption spectrometer (Graf et al., 2018; Tuzson et al., 2020) capable of measuring atmospheric methane mole fractions at $1\,\mathrm{s}$ time resolution. The instrument achieves a precision $(1\sigma)$ of $1.1\,\mathrm{ppb}$ at $1\,\mathrm{s}$ and $0.1\,\mathrm{ppb}$ at $100\,\mathrm{s}$ averaging time. This performance is mainly preserved also under flight conditions. The analyzer has a compact footprint $(15 \times 45 \times 25\,\mathrm{cm^3})$ and weighs only $2.1\,\mathrm{kg}$, including batteries.

        The analyzer uses a distributed feedback (DFB) quantum cascade laser (QCL) emitting in the mid-infrared at $7.83\,\mathrm{\mu m}$.

During the flight, air flows passively through an open circular absorption cell of $77\,\mathrm{mm}$ radius. Multiple reflections of the laser beam on the segmented inner surface results in an effective optical path of about $10\,\mathrm{m}$. The compact design of the multipass cell



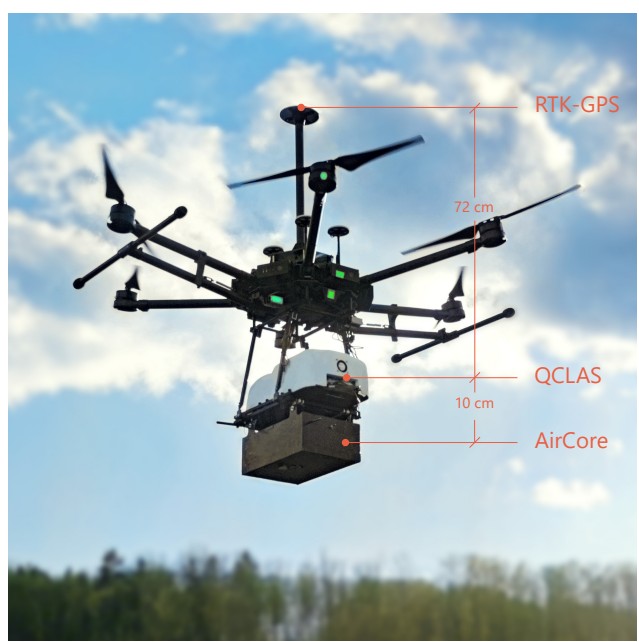

**Figure 1.** The embedded UAV system used for $CH_4$ detection: the QCLAS analyzer and the active AirCore sampling system are mounted below a Matrice 600 DJI hexacopter equipped with a RTK-GPS system.

combines the advantage of a long optical path with mechanical stability allowing efficient and interference-free beam folding (Graf et al., 2018).

The energy consumption of the spectrometer has been minimized using a customized System-on-Chip (SoC) FPGA-based hardware control and data acquisition as well as a custom-made laser driving electronics (Liu et al., 2018). The instrument's precision, linearity, and its dependence on various environmental parameters were characterized and consequently validated under field conditions (Tuzson et al., 2020).

Real-time data synchronization between the instrument and a computer on the ground is made possible by a wireless bi-directional data link (SkyHopper PRO). This allows real-time access to the raw spectra and all hardware parameters during the flights, which enables the operator to do real-time spectral fitting and logging. Thus, the operator is provided with full control of the hardware, continuous monitoring of the instrument's status, as well as in-situ monitoring of the ambient $CH_4$ values during the flights.

## 2.2 Active AirCore

The active AirCore, designed for atmospheric sampling on a UAV, consists of $50\,\mathrm{m}$ thin-wall stainless-steel tubing, a dryer, a micro-pump, and a data-logger (Andersen et al., 2018). The whole system is enclosed in a carbon fiber box with a compact footprint ($1.1\,\mathrm{kg}$, $34 \times 19.5 \times 12.0\,\mathrm{cm}^3$) making it suitable for drone-based measurements.





Prior to each quantification flight, the active AirCore is flushed with a calibrated fill gas, spiked with about $10\,\mathrm{ppm}$ CO, in order to identify the starting point of ambient air sampling. Shortly before the integrated drone system takes off, the micro-pump is turned on to sample ambient air and immediately after the quantification flight, it is turned off to stop sampling ambient

air. The active AirCore samples are then consequently analyzed on site with a trace gas analyzer (CRDS G2401-m, Picarro, Inc., CA, USA).

## 2.3 RTK-GPS System

Readily available commercial drones, including the Matrice 600 DJI, rely on simple global positioning systems (GPS), similar to systems found in other utilities such as mobile phones and smart watches. GPS readings combined with ambient pressure

measurements are used to obtain the spatial coordinates, specifically the altitude, of the drone at the time of flight. Manufacturer specification reports vertical accuracy of this type of drones to be $\pm 0.50\,\mathrm{m}$. However, this level of accuracy is not sufficient for our purpose that requires a precise spatial mapping of the plumes, especially with respect to height.

Alternatively, real-time kinematic (RTK) positioning can be employed to enhance positioning accuracy. Nowadays, accuracy at the level of $\mathrm{cm}$ are possible even with low-cost receivers (such as the NEO-M8P) by capturing raw streams from the GPS

satellites and then post processing the logs with open source programs (e.g. RTKLIB). For our purpose, we deployed two RTK-GPS boards from Spark-Fun. The rover was integrated with the data acquisition of the drone-based QCLAS system, while the second board was deployed as a stand-alone, battery-powered base station. Post-processing of raw-data was done using RTKLIB, which returns corrected coordinates.

A direct comparison of an altitude time-series between the UAV-GPS and the RTK-GPS data in one of our flights is presented

in Fig. 2. Quantified average drift of the UAV-GPS for the entire duration of the controlled-release experiment was found to be $0.1\,\mathrm{cm\,s^{-1}}$, equivalent to $0.6\,\mathrm{m}$ of altitude drift for a $10\,\mathrm{min}$ duration measurement flight. Details on how this altitude drift affects our quantification estimate are discussed in Sect. 5

## 3 Tracer Release Experiment

The release experiment was performed over a managed agricultural field (Agrar Hauser) near the city of Dübendorf, Switzer-

land. The field is a seasonal cropland with an access road mainly used by pedestrians and bikers. The location is relatively flat, but is shielded by a forested hill about $250\,\mathrm{m}$ in the south. The release experiment was performed from 23 February to 14 March 2020 with a total of 9 days of active measurements. There is no livestock or other significant methane source in the vicinity of the field, making it an ideal location for the experiment. The selection of active days was mainly based on favorable weather conditions, i.e. days with no precipitation and with sufficient but not too strong winds. Local wind speeds during the

selected days ranged from $1 - 7\,\mathrm{m\,s^{-1}}$. A total of 35 measurement flights were performed during the whole campaign, out of which 18 are suitable for quantification. The rest had to be discarded mainly due to technical problems with either the drone, the analyzers, the tracer release, or the GPS device. A sample measurement flight is presented in Fig. 3, which also provides an aerial view of the site.





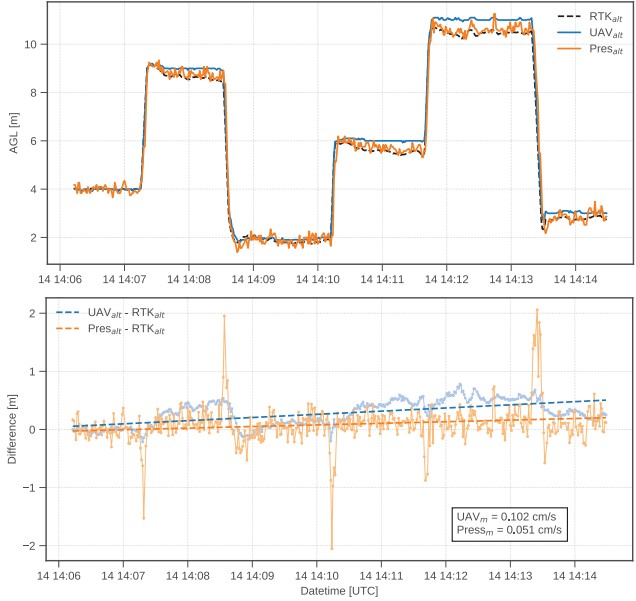

**Figure 2.** Recorded altitude during the flight with code 314_03. Color coded lines represent the altitude measured using three different systems. The black dashed line and blue line corresponds to the altitude recorded by RTK and UAV-GPS, respectively. The orange line refers to the altitude derived using the pressure sensor.

Alongside the drone flights, a second quantification method based on stationary measurements with an independent methane analyzer was applied on the first three days of the campaign for comparison. The method, called OTM-33A (U.S. EPA, 2014), is presented in more detail in Sect. 4.2. In order to avoid any possible bias in the data processing towards the real tracer release, two of the releases were conduced as blind experiments, where a third party person released methane at a rate not known to the team.

An artificial methane source, in the form of natural gas of which $92.2\%$ is $CH_4$, was released from a $50\,L$ high-pressure cylinder. The gas was directed through a $100\,m$ long $1.2\,cm$ inner-diameter tubing to the release point. The end of the tubing was placed at about $1.5\,m$ above surface. A mass flow controller (MFC, red-y series, Vögtlin Instruments) calibrated for methane up to $100\,L\,min^{-1}$ at normal conditions was used to regulate the gas release. A summary of the release rates during the experiment is given in Table 1. At the start of each measurement day, a suitable location of the release was determined based on prevailing winds. Meteorological conditions were measured using 3D and/or 2.5D anemometers, which were usually placed next to the release point of the source.



**Table 1.** Overview of MethAne Tracer Release EXperiment (MATRIX).

| Date | Flight Code | Time [UTC] | Release rate [gs⁻¹] | Downwind dist. [m] | WS [ms⁻¹] | WD [deg. from N] | Stab. | Instruments Present |
|---|---|---|---|---|---|---|---|---|
| 23-Feb | 223_01 | 13:26:03 - 13:43:05 | $0.48 \pm 0.04$ | 42 | $4.98 \pm 1.41$ | $277 \pm 18$ | N | O, Q |
| 24-Feb | 224_01 | 15:40:50 - 15:48:01 | $0.29 \pm 0.03$ | 94 | $5.21 \pm 1.61$ | $283 \pm 15$ | N | O, Q |
| 25-Feb | 225_01 | 10:30:00 - 10:40:41 | $0.29 \pm 0.03$ | 50 | $4.53 \pm 1.25$ | $304 \pm 11$ | N | A, O, Q |
| | 225_02 | 10:50:17 - 11:01:45 | $0.29 \pm 0.03$ | 48 | $5.68 \pm 1.18$ | $304 \pm 14$ | N | A, O, Q |
| | 225_03 | 11:16:50 - 11:24:23 | $0.29 \pm 0.03$ | 45 | $6.08 \pm 1.49$ | $304 \pm 12$ | N | A, O, Q |
| 08-Mar | 308_02 | 13:17:37 - 13:28:26 | $0.26 \pm 0.02$ | 40 | $1.69 \pm 0.76$ | $271 \pm 19$ | U | A, Q, R |
| 09-Mar | 309_01 | 09:19:02 - 09:28:54 | $0.29 \pm 0.03$ | 18 | $2.61 \pm 1.31$ | $284 \pm 28$ | N | A, Q |
| | 309_02 | 09:52:08 - 10:03:28 | $0.29 \pm 0.03$ | 31 | $2.65 \pm 1.06$ | $284 \pm 28$ | N | A, Q, R |
| 12-Mar | 312_01 | 14:11:00 - 14:21:07 | $0.31 \pm 0.03$ | 46 | $3.49 \pm 0.83$ | $312 \pm 11$ | N | A, Q, R |
| | 312_03 | 14:58:47 - 15:09:39 | $0.39 \pm 0.03$ | 77 | $3.55 \pm 0.71$ | $306 \pm 13$ | N | A, Q, R |
| 13-Mar | 313_01 | 11:36:05 - 11:44:02 | blind | 51 | $3.29 \pm 0.97$ | $284 \pm 18$ | U | A, Q, R |
| | 313_02 | 11:57:58 - 12:07:15 | blind | 50 | $2.88 \pm 1.03$ | $282 \pm 16$ | U | A, Q, R |
| | 313_03 | 13:33:14 - 13:40:44 | $0.46 \pm 0.04$ | 129 | $2.34 \pm 1.07$ | $257 \pm 32$ | U | A, Q, R |
| | 313_04 | 13:51:32 - 14:02:45 | $0.48 \pm 0.04$ | 136 | $2.63 \pm 0.82$ | $282 \pm 46$ | U | A, Q, R |
| | 313_05 | 14:16:07 - 14:27:06 | $0.52 \pm 0.05$ | 102 | $2.15 \pm 0.71$ | $280 \pm 46$ | U | Q, R |
| 14-Mar | 314_01 | 12:40:41 - 12:49:17 | $0.26 \pm 0.02$ | 40 | $0.72 \pm 0.33$ | $111 \pm 39$ | U | A, Q, R |
| | 314_02 | 13:01:25 - 13:13:49 | $0.44 \pm 0.04$ | 40 | $0.51 \pm 0.22$ | $180 \pm 44$ | U | A, Q, R |
| | 314_03 | 14:06:00 - 14:13:49 | $0.68 \pm 0.03$ | 44 | $0.63 \pm 0.27$ | $154 \pm 37$ | U | A, Q, R |

Instruments - A: AirCore, Q: QCLAS, O: OTM-33A, R: RTK | Meteorological stability - N: Neutral, U: Unstable, S: Stable

## 4 Method

### 4.1 Mass-Balance

Mass-balance methods have been applied extensively to aircraft-based measurements for quantifying emissions from facility scale (e.g. Ryerson et al., 2001; Karion et al., 2013; Gordon et al., 2015; Lavoie et al., 2015; Tadić et al., 2017) up to urban and regional scale (e.g. Cambaliza et al., 2015; Pitt et al., 2019; Fiehn et al., 2020; Klausner et al., 2020). The quantification involves flying downwind and/or around a region of interest at a single vertical height or multiple heights. Emission rates are quantified by taking the net difference between fluxes into and out of a volume containing the source. Subtracting a large-scale background, the inflow is usually assumed to be zero and the outflow is determined from the enhancements above background inside the plume downwind of the source together with measurements or model assumptions of wind speed. With the advent of UAVs, estimating emissions using the cross-sectional mass-balance method originally used for aircraft may be adapted to smaller scale and more localized sources. Emission quantification is best performed by flying the UAV downwind of a given





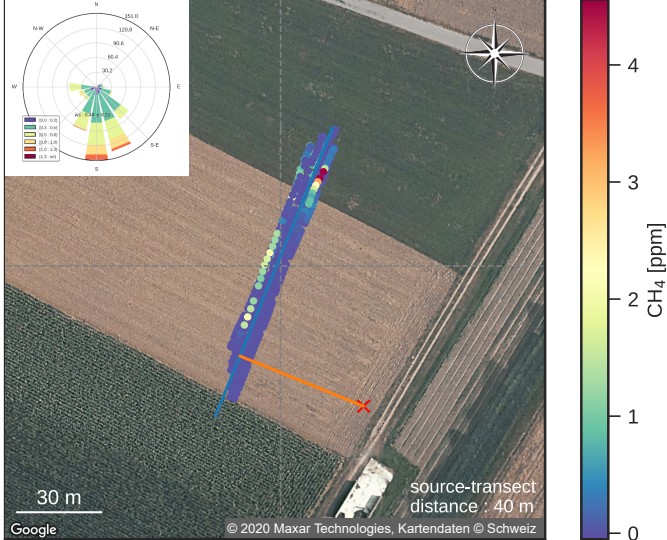

**Figure 3.** Measured methane mole fraction during MATRIX with flight code 314_02. The red cross indicates the location of the artificial source. The source-transect distance is computed as the perpendicular distance between the source and the measurement plane. The flight trajectories are illustrated as colored dots indicating the measured local $CH_4$ concentrations. Wind and turbulence conditions are measured with a 3D sonic anemometer located next to the source.

source perpendicular to the main wind direction at multiple altitudes above ground up to an altitude, $z_{max}$, where no discernible change in methane mixing ratio is observed. Background mole fractions can be determined from measurements outside of the plume or from measurements upwind of the source.

Applying mass conservation for a chemically non-reactive gas within a control volume, the emission flux downwind of a given source can be quantified as:

$$Q_c = \int_{y_{min}}^{y_{max}} \int_0^{z_{max}} c(y,z)\boldsymbol{u}(y,z) \cdot \hat{\boldsymbol{n}} dz dy \tag{1}$$

where $Q_c$ is the sum of methane emission fluxes within the area of interest. The y-axis is aligned with the vertical cross-section in which the UAV is flown. The integral over this two-dimensional plane is approximated in the observations as a discrete
summation of the product of the mass concentration of methane above background $c(y,z)$ and the component of the horizontal wind vector $\boldsymbol{u}(y,z)$ normal to the vertical cross-section, i.e. parallel to the unit vector $\hat{\boldsymbol{n}}$. In doing so, it is assumed that there are no other significant sources of methane emissions upwind besides the controlled release.

### 4.2 OTM-33A

Other test methods (OTM) 33A was introduced by U.S. EPA (2014) to quantify emissions from natural oil and gas sites
emitting at near ground level without having the need to access the site. This approach heavily relies on the assumption that





plume dispersion is governed by point source Gaussian (PSG), and thus requires certain conditions to be met for effective quantification. In particular, the target source must come from a single point and no nearby sources should contribute to the measurement. Furthermore, no obstacle should be present between the source and the measurement point. Lastly, measurements of methane and meteorological parameters should be collected at $1-2\,\mathrm{Hz}$ and should be taken under rather steady wind

conditions with a wind speed of at least $1\,\mathrm{m\,s^{-1}}$ blowing consistently from the source to the measurement point over a period of at least $15-20$ minutes.

The emission rate, $Q_\mathrm{c}$, of the point source is then estimated using the following equation that is based on spatial integration over a Gaussian shaped plume of horizontal width $\sigma_y$ and vertical width $\sigma_z$

$$Q_\mathrm{c} = 2 \cdot \pi \cdot \sigma_y \cdot \sigma_z \cdot \overline{U} \cdot\ C_\mathrm{peak} \tag{2}$$

The horizontal and vertical dispersion coefficients $\sigma_y$ and $\sigma_z$ are parameterized as a function of distance from the source using a lookup table developed by (U.S. EPA, 2014) based on Pasquill stability classes. The average wind speed during the measurements is $\overline{U}$, and the $C_\mathrm{peak}$ is obtained by taking the peak of a Gaussian fit of methane enhancements with respect to wind directions, binned into $10°$.

The method was characterized using controlled-release experiments (Brantley et al., 2014; Robertson et al., 2017; Edie et al.,

2020), which suggested that the method has a $2\sigma$ error of $\pm70\,\%$ with a slight negative bias of about $5\,\%$ (Heltzel et al., 2020). It was eventually used to quantify emissions from oil and gas plants in the US (Brantley et al., 2014; Robertson et al., 2017) and results were compared to direct measurements simultaneously performed on site (Bell et al., 2017). Quantification estimates from Robertson et al. (2017) were slightly lower as compared to direct measurements, but most emissions were captured within the $2\sigma$ uncertainty. A further analysis of controlled-release data by Edie et al. (2020) suggested that the error caused

by variations in wind speed, number of sources, and release height is small compared to the method's uncertainty, and has no significant effect on the accuracy of the emission estimates. This implies that the method is also applicable under conditions outside of the strict bounds of the original formulation by U.S. EPA (2014).

The OTM-33A method was applied alongside measurement flights on the first three days of the MATRIX campaign. Prior to quantification, the dominant wind direction was chosen following screening recommendations of U.S. EPA (2014). Once

determined, a portable $CH_4$ analyzer (LI-7810, LI-COR, Inc.) and a 3D sonic anemometer (uSonic-3 Scientific, METEK) were placed in a stationary position, $35-70\mathrm{m}$ downwind of the source, to measure continuous methane mole fractions and meteorological parameters at $1\,\mathrm{Hz}$ with an inlet height set at $2.5\,\mathrm{m}$ above ground.

### 4.3  Estimation of wind speeds along the drone flight

Local meteorological conditions were measured using the 3D sonic anemometer placed next to the artificial point source

sampling at an altitude of $2\,\mathrm{m}$ above ground. The anemometer has a sampling rate of $20\,\mathrm{Hz}$, and measurements were averaged every second. Wind speeds were then decomposed into components normal and parallel to the measurement plane. Turbulence parameters such as friction velocity $u^*$ and Obukhov length $L$ were computed for each measurement flight. In this study, three different ways of computing the normal wind component along the drone transects were tested. The first and most simple



approach was to apply the mean normal component of the wind vector $\boldsymbol{u}$ measured during the whole flight uniformly to all

points in the vertical cross-section. Eq. (1) can then be simplified to

$$Q_c = U \int\limits_{y_{\min}}^{y_{\max}} \int\limits_{0}^{z_{\max}} c(y,z) dz dy \tag{3}$$

where $U$ is the mean of the normal component of the wind.

A second approach involved the construction of a theoretical logarithmic wind profile to vertically extrapolate the measure-

ments at $2\,\mathrm{m}$ to the whole altitude range covered by the drone. The stability condition of the atmospheric surface layer was

determined using the Obukhov length. Depending on whether the surface layer was neutral, stable, or unstable, the roughness

length, $z_0$, was derived using the logarithmic profile

$$\overline{u}_z = \frac{u_*}{\kappa} \left[ \ln\left(\frac{z}{z_0}\right) - \Psi_m\left(\frac{z}{L}\right) \right] \tag{4}$$

where $\overline{u}_z$ is the normal component of the wind vector at the height of the actual measurement $z$ and $\Psi_m$ is a profile function

depending on the stability of the atmosphere. Following Högström (1988), we applied the following structure functions

$$\Psi_m = \begin{cases} 0 & \text{neutral} \\ -6\frac{z}{L} & \text{stable} \\ 2\ln\left(\frac{1+x}{2}\right) + \ln\left(\frac{1+x^2}{2}\right) - 2\arctan x + \frac{\pi}{2} & \text{unstable} \end{cases}$$

with $x = (1 - 15z/L)^{0.25}$. Instead of using a constant wind at all levels as in the first approach, the wind speed thus varied with

altitude.

The third approach involved taking the $1\,\mathrm{s}$-average normal wind component and projecting it onto the measurement plane

by matching the timestamp of the anemometer to the to GPS location of the UAV during the time of measurement. This

allows accounting for changes in wind conditions over the period of a drone flight. The measurement plane is assumed to

be sufficiently close to the anemometer that the wind measurements are representative for the conditions encountered by the

drone. With a typical downwind distance of about $40\,\mathrm{m}$ and a wind speed of $4\,\mathrm{m\,s^{-1}}$ (see Table 1), a wind gust measured at

the anemometer would arrive at the measurement plane after only $10\,\mathrm{s}$. After projecting each normal wind component to the

location of the drone, a wind field is constructed by ordinary kriging using the projected wind data.

**4.4    Post-processing of drone measurements**

Timestamps of the $CH_4$ data reported by the QCLAS and positional coordinates from the RTK-GPS system were synchronized

by performing a cross-correlation between the longitude and latitude reading of the built-in GPS of the QCLAS and the RTK-

GPS system. After determining the delay between clocks, timestamps from the QCLAS were shifted to match the RTK-GPS

system, which is considered to be the real time that all other clocks in the system follow.

Background $CH_4$ mole fractions were removed from the data set by using the Robust Extraction Baseline Signal (REBS)

algorithm developed by Ruckstuhl et al. (2012). Take-off and landing times of the UAV are noted and all data before and after

the flight are removed.





### 4.4.1 Processing of Active AirCore measurements

In contrast to the $CH_4$ mole fractions measured by the fast-response QCLAS analyzer, characterized by sharp and instantaneous

elevations, the measurements by the AirCore resulted in a rather smooth signal, as presented in Fig. 4. Instantaneous methane plumes usually did not have a Gaussian shape, but rather showed complex structures with small patches of elevated concentrations due the chaotic nature of turbulence. These sharp concentration gradients were fully captured by the fast-response QCLAS, but were smeared out by the AirCore system, which has a much slower response due to mixing in the sampling tube and later in the CRDS analyzer.

To determine the magnitude of smoothing present in the AirCore measurements, we flew the two instruments simultaneously with the drone, while measuring the same point source downwind, as shown in Fig. 1. We then transformed the fast-response QCLAS measurements to mimic the smooth and smeared out AirCore data. Adapting the in-flight spectral calibration algorithm of imaging spectrometers developed by Kuhlmann et al. (2016) into a smoothing, shifting, and stretching (3S) algorithm, we obtained the parameters applied in transforming the QCLAS measurements to match the measurements from the AirCore.

This implied convoluting the QCLAS data with a non-linear fit function, and subsequently shifting and stretching the convoluted QCLAS measurements. The smoothing of the AirCore measurements is dominated by the response of the CRDS analyzer, i.e. air mixing in the analyzer cavity (Andersen et al., in review; Vinkovich et al., in prep), but also influenced by molecular diffusion during sample storage as well as Taylor diffusion during sampling and analysis (Karion et al., 2010). Analogous to Kuhlmann et al. (2016), we approximate the active AirCore measurement as a vector $\mathbf{y}$ defined as

$$\mathbf{y} = \mathbf{F}(\mathbf{x}, \mathbf{b}) + \mathbf{e} \qquad (5)$$

where $\mathbf{F}$ is a function with a Gaussian form that depends on the state vector $\mathbf{x}$ and parameter vector $\mathbf{b}$. The state vector $\mathbf{x}$ contains three elements, where the first two are the control points describing the shift and stretch parameter and the third as the smoothing parameter. The error vector $\mathbf{e}$ represents the instrument's error as well as the error from the Gaussian fit.

Given a measurement vector $\mathbf{y}$, the goal is to find an optimal state vector $\hat{\mathbf{x}}$ that minimizes the merit function:

$$\chi^2(\mathbf{x}) = (\mathbf{y} - \mathbf{F}(\mathbf{x}))^\top \mathbf{S}_\epsilon^{-1} (\mathbf{y} - \mathbf{F}(\mathbf{x})) \qquad (6)$$

where $\mathbf{S}_\epsilon$ is the measurement covariance matrix. The optimal state vector $\hat{\mathbf{x}}$ is calculated iteratively using the Gauss-Newton method:

$$\mathbf{x}_{i+1} = \mathbf{x}_i + \hat{\mathbf{S}}_i (\mathbf{K}_i^\top \mathbf{S}_\epsilon^{-1} (\mathbf{y} - \mathbf{F}(\mathbf{x}_i, \mathbf{b})) \qquad (7)$$

The $\hat{\mathbf{S}}_\mathbf{i}$ is the *a posteriori* error covariance matrix

$$\hat{\mathbf{S}}_\mathbf{i} = \left( \mathbf{K}_i^\top \mathbf{S}_\epsilon^{-1} \mathbf{K}_i \right)^{-1} \qquad (8)$$

where $\mathbf{K}_i$ is a Jacobian matrix whose elements are the partial derivatives of the forward model with respect to the state vector, and the iteration stops if:

$$(\mathbf{x}_i - \mathbf{x}_{i+1})^\top \hat{\mathbf{S}}_i^{-1} (\mathbf{x}_i - \mathbf{x}_{i+1}) < n \qquad (9)$$





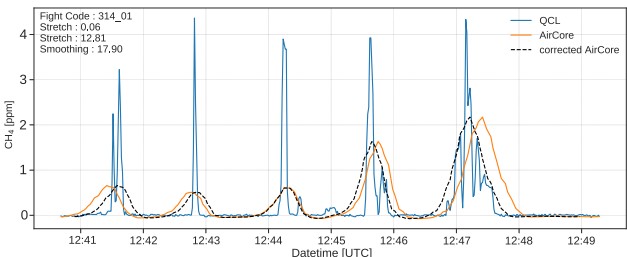

**Figure 4.** Methane mole fraction time-series obtained by simultaneously flying the active AirCore system (orange line) and the in-situ QCLAS analyzer (blue line). Black dashed line represents the corrected AirCore measurements by using the shifting and stretching parameter obtained from the 3S algorithm.

where $n = 3$ is the number of state vector elements.

## 4.5 Cluster based kriging

In order to compute the flux through the vertical cross-section, the spatially discrete samples were interpolated to fill all gaps in the plane. Kriging is a popular method of stochastic interpolation in which the produced interpolated surface is modelled by a Gaussian process governed by prior covariance kernels, which is a realization of many possible outcomes that could have produced the known data points.

Kriging models have been widely used in atmospheric science and air quality as a tool for data analysis and prediction (e.g. Wong et al., 2004; Tadić et al., 2015; Michael et al., 2019; Tadić et al., 2015). However, applying kriging to airborne measurements is faced with several challenges. Standard ordinary kriging assumes spatial stationarity of the geophysical field (Tadić et al., 2015) and all data points are assumed to be taken from a unimodal single probability distribution. Both assumptions are not necessarily true when a temporally varying plume is sampled sequentially over the duration of a flight. Furthermore, the scales of spatial variability of methane inside the plume and in the background are largely different, which violates the assumption of a unimodal distribution.

In order to overcome these issues, a cluster-based kriging (van Stein et al., 2020) was adapted. The process may be summarized into three main steps: i) Partitioning the data-set into smaller clusters; ii) Training an adequate kriging model for each cluster; iii) Combining all kriging models to predict values (i.e., methane mole fractions) at unknown locations.

## 4.5.1 Data clustering

Cluster analysis or clustering is a process of grouping data into subsets according to a degree of similarity found inherently within the data. Clustering can be performed in many ways and can generally be divided into two basic types, hard- and soft-clustering.

Hard clustering is achieved when the data is split into smaller disjoint data-sets, and the resulting label of a data-point belongs to one and only cluster. The most common example of an algorithm that implements hard clustering is *K-means*. On





the other hand, soft clustering splits the data into smaller data-sets with small overlaps, and returns a probability of how much a data-point is associated with a specific cluster. A soft clustering approach is favored in this study as this approach increases the final model accuracy (van Stein et al., 2020). One of the widely used models to perform soft clustering is a Gaussian Mixture Model (GMM) (Reynolds, 2015). GMM is the type of model that will be used here.

Given a set $\mathcal{X} = \{(\boldsymbol{x}^1, y^1), \ldots, (\boldsymbol{x}^n, y^n)\}$ of methane mole fractions $y^i$ acquired at locations $\boldsymbol{x}^i$ for $i = \{1, \ldots, n\}$, where $n$ is the number of data points collected, the goal is to split the input data $\mathcal{X}$ into a set $\mathcal{S}$ composed of several Gaussian components $k$, such that:

$$\mathcal{S} = \{\mathcal{X}_1, \ldots, \mathcal{X}_k\}, \quad \text{where} \bigcup_{j=1}^{k} \mathcal{X}_j = \mathcal{X}. \tag{10}$$

Each cluster $\mathcal{X}_j$ in the set $\mathcal{S}$ is assumed to have a Gaussian shape in three dimensions, namely the 2D spatial location $\boldsymbol{x}$ and
methane mole fraction $y$. The shape of each $\mathcal{X}_j$ being determined by a set of parameters $\theta_j = \{\pi_j, \mu_j, \Sigma_j\}$ where $\pi_j$ is the mixing probability, $\mu_j$ is the mean, and $\Sigma_j$ is the covariance (i.e., spread) of the Gaussian. Each cluster $\mathcal{X}_j$ is acting together to model the overall density of $\mathcal{X}$. The probability distribution of $\mathcal{X}$ given a global mixture model parameter $\theta = \{\theta_1, \ldots, \theta_k\}$ is defined as:

$$p(\mathcal{X} \mid \theta) = \sum_{j=1}^{k} \pi_j \mathcal{N}(\mathcal{X}_j \mid \mu_j, \Sigma_j), \quad \text{where} \sum_{j=1}^{k} \pi_j = 1 \tag{11}$$

$\mathcal{N}$ is the normal distribution with mean $\mu_j$ and width $\Sigma_j$. The global mixture model parameter $\theta$ that best describes the data must be learned. The most established method to learn this parameter is through the use of an expectation-maximization (EM) algorithm. Given an initial parameter $\theta$, the EM algorithm aims to estimate a new $\bar{\theta}$, such that $p(\mathcal{X} \mid \bar{\theta}) \geq p(\mathcal{X} \mid \theta)$. The new parameter then becomes the old parameter for the next iteration, and this process is repeated until a convergence threshold is satisfied. The *a posteriori* probability of a data point $(\boldsymbol{x}^i, y^i)$ belonging to cluster $\mathcal{X}_j$ with parameters $\theta_j$ is then given by:

$$\text{Pr}((\boldsymbol{x}^i, y^i) \in \mathcal{X}_j \mid \theta_j) = \frac{\pi_j \mathcal{N}(\mathcal{X}_j \mid \mu_j, \Sigma_j)}{\sum_{j=1}^{k} \pi_j \mathcal{N}(\mathcal{X}_j \mid \mu_j, \Sigma_j)}$$

$$\text{for } j = 1, \ldots, k \tag{12}$$

Once the model parameter $\theta$ and the membership probability of a data point belonging to a cluster is learned, the clustered data points are expanded to the whole domain and the membership probability of an unobserved location point $\boldsymbol{x}_j^t$ belonging to cluster $\mathcal{X}_j$ is computed as well.

For typical trace gas distribution modelling, Stachniss et al. (2009) suggested to use a mixture of only two clusters. The first
cluster corresponds to measured background mole fractions, whereas the second cluster corresponds to elevated measurements. This choice is motivated by the fact that the spatial scales of variability are largely different between the two clusters. Tests with larger mixtures applied to our data-set showed that a two-cluster mixture is indeed sufficient to achieve good results.





### 4.5.2 Kriging estimate

Once the data-set has been clustered and the membership probability of each data-point belonging to a cluster has been com-
puted, ordinary kriging models are trained for each cluster separately to spatially interpolate the field of interest. Since data
points for ordinary kriging can only belong to one of the two clusters, the kriging model for each cluster is learned using hard
clustered data-points, either belonging to the background or the elevated cluster. Interpolation of a geophysical field from a
spatially sparse data-set is highly dependent on a chosen covariance kernel $K$, which statistically describes the relationship
between two spatial points using a set of hyper-parameters $\lambda = \{l, \sigma\}$, where $l$ refers to the length-scale and $\sigma^2$ is the overall
variance (i.e., noise) coming from the data. There are several ways to define the covariance kernel. In this study, the Matèrn
5/2 covariance kernel is chosen as it performs better compared to other frequently used kernels, such as a squared exponential
function as shown by Stachniss et al. (2009). In their study, they established that the Matèrn covariance kernel has a lesser
degree of smoothing compared to other kernels, which resembles more closely the nature of gas distributions in the vicinity of
a localized source. Optimizing the hyper-parameters of the covariance kernel $K$ for each cluster is done by evaluating a log-
marginal-likelihood (LML) using a set of initial parameters, which are increased or decreased incrementally until a maximum
value is obtained. The whole process of clustering the data-set into two clusters followed by optimizing the hyper-parameters
of each cluster was implemented using the scikit-learn package of python.

Optimized hyper-parameters $\lambda_j = \{l_j, \sigma_j\}$ for each cluster $\mathcal{X}_j$ are used to perform ordinary kriging to predict a data-point
$(\boldsymbol{x}_j^t, y_j^t)$ of unmeasured methane mole fraction $y_j^t$ at an unobserved location $\boldsymbol{x}_j^t$. The resulting interpolated field from kriging
is a Gaussian distribution $\mathcal{N}$ expressed as

$$y_j^t \mid \mathcal{X}_j \sim \mathcal{N}\left(m_j(\boldsymbol{x}^t), s_j^2(\boldsymbol{x}^t)\right) \tag{13}$$

with mean $m_j$ and variance $s_j^2$.

The final predicted value $(\boldsymbol{x}^t, y^t)$ of methane mole fractions $y^t$ at each point $(x^t, y^t)$ is obtained by combining the results of
all the kriging models together $(\boldsymbol{x}_j^t, y_j^t)$ with the respective membership probability of each spatial point $\boldsymbol{x}^t$ used as weights
$w_j$, denoted as

$$w_j = \Pr\left(C = j \mid \mathcal{X}, \boldsymbol{x}^t\right), \quad \text{for } j = 1, \ldots, k \tag{14}$$

where $C$ is the cluster indicator ranging from 1 to $k$.

Thus, the expected value of methane mole fraction $y^t$ at each spatial point $\boldsymbol{x}^t$ is

$$\mathrm{E}\left[y^t \mid \mathcal{X}, y, \mathbf{x}^t\right] = \sum_{j=1}^{k} w_j m_j(\boldsymbol{x}^t) \tag{15}$$

and the variance of the expected value is (van Stein et al., 2020)





$$\text{Var}\left[y^t \mid \mathcal{X}, y, x^t\right]$$

$$= \sum_{j=1}^{k} w_j \left(s_j^2 \left(\boldsymbol{x}^t\right) + m_j^2 \left(\boldsymbol{x}^t\right)\right) - \left(\sum_{j=1}^{k} w_j m_j(\boldsymbol{x}^t)\right)^2 \tag{16}$$

## 5 Results and Discussion

### 5.1 Example of quantification procedure

An illustration of the clustering and kriging approach used to map a discrete set of data points onto the whole measurement plane is presented in Fig. 5 for flight 312_03 on 12 Mar 2020. The time-series presented in the upper left in panel (a) was first mapped onto the 2D measurement plane composed of horizontal distance and vertical altitude. The time-series, composed as a set of ordered spatial and methane concentration points $(\mathbf{x}, y)$, was then fed into a GMM to partition the data-set into two clusters, namely, the background and the elevated cluster. The GMM returns the membership probability of a data point belonging to one cluster or the other. The membership probability of each data point was then expanded to the whole domain

to unobserved locations as shown in Fig. 5b. In a next step, ordinary kriging was applied to each cluster separately to produce a background and an elevated $CH_4$ distribution, respectively (Fig. 5b left panels). Finally, the kriging results for each cluster were combined with their respective membership probability. The resulting kriging field is illustrated in Fig. 5c with the expected value computed according to Eq. (15) and prediction uncertainty, i.e., the square root of variance according to Eq. (16). A reconstructed time-series of predicted methane mole fraction was compared to the original time-series of measured methane

is shown in Fig. S1. The peaks of predicted methane mole fraction are lower but broader compared to the original methane time-series as expected as kriging applies smoothing in the data.

The measured average ambient air temperature $T$ [K] and pressure $p$ [Pa] during the flight was used to convert the obtained kriging field of methane mole fraction $\chi_{CH_4}$ [ppm] into concentrations $\rho_{CH_4}$ [g m$^{-3}$]:

$$\rho_{CH_4} = \chi_{CH_4} \frac{p M_{CH_4}}{RT} \tag{17}$$

where $R$ is the gas constant ($8.3144\,\text{J}\,\text{K}^{-1}\,\text{mol}^{-1}$) and $M_{CH_4}$ is the molar mass of methane ($16.04\,\text{kg}\,\text{mol}^{-1}$). The influence of humidity, which introduces an error of no more than $1\,\%$, was ignored in this equation. As we are only interested in methane elevations above the background, this uncertainty is considered small.

The concentration field was combined with wind fields using three different wind treatments as discussed in Sect. 4. Finally,

an emission rate $Q_c$ was estimated as a scalar (dot) product of the concentration field $\mathbf{C}$ and the wind field $\mathbf{U}$ written as vectors:

$$Q_c = \left(\mathbf{C}^\top \cdot \mathbf{U}\right) \Delta y \Delta z \tag{18}$$

where $\Delta y$ and $\Delta z$ are the regularly spaced intervals in the horizontal and vertical direction.





The emission rate $Q_C\left(\mathbf{C}, \mathbf{U}\right)$ is a function of two variables $\mathbf{C}$ and $\mathbf{U}$ and the overall error propagation of the function is:

$$\Delta Q_c^2 = \left(\frac{\partial Q_c}{\partial \mathbf{C}}\Delta \mathbf{U}\right)^2 + \left(\frac{\partial Q_c}{\partial \mathbf{U}}\Delta \mathbf{C}\right)^2 \Delta y^2 \Delta z^2 \tag{19}$$

The concentration field $\mathbf{C}$ and wind field $\mathbf{U}$ come with their respective covariance matrix $\mathbf{K}_\mathrm{C}$ and $\mathbf{K}_\mathrm{U}$ provided by kriging, and the above equation becomes:

$$\Delta Q_c^2 = \left(\mathbf{U}^\top \cdot \mathbf{K}_\mathrm{C} \cdot \mathbf{U} + \mathbf{C}^\top \cdot \mathbf{K}_\mathrm{U} \cdot \mathbf{C}\right)\Delta y^2 \Delta z^2 \tag{20}$$

In cases where $\mathbf{U}$ is a scalar constant or logarithmic profile, the uncertainty of the wind is estimated by computing the standard deviation ($1\sigma$) of the mean wind speed normal to the measurement plane during the flight.

Cluster-based kriging produces the concentration field $\mathbf{C}$ as a linear combination of two distinct concentration fields $\mathbf{C}_\mathrm{elev}$ and $\mathbf{C}_\mathrm{bg}$ with weights $\mathbf{w}_\mathrm{elev}$ and $\mathbf{w}_\mathrm{bg}$

$$\mathbf{C} = \mathbf{w}_\mathrm{elev}\mathbf{C}_\mathrm{elev} + \mathbf{w}_\mathrm{bg}\mathbf{C}_\mathrm{bg} \tag{21}$$

where $\mathbf{w}_\mathrm{elev}$ and $\mathbf{w}_\mathrm{bg}$ are vectors of the same length as $\mathbf{C}_\mathrm{elev}$ and $\mathbf{C}_\mathrm{bg}$. Both concentration fields come with a covariance matrix $\mathbf{K}_\mathrm{Celev}$ and $\mathbf{K}_\mathrm{Cbg}$ as determined by kriging. The weights $\mathbf{w}_\mathrm{elev}$ and $\mathbf{w}_\mathrm{bg}$ are constants without uncertainties.

The concentration field $\mathbf{C}\left(\mathbf{C}_\mathrm{elev}, \mathbf{C}_\mathrm{bg}\right)$ is a function of two variables $\mathbf{C}_\mathrm{elev}$ and $\mathbf{C}_\mathrm{bg}$ and the error propagation of the function is:

$$\Delta \mathbf{C}^2 = \left(\frac{\partial C}{\partial \mathbf{C}_\mathrm{elev}}\Delta \mathbf{C}_\mathrm{elev}\right)^2 + \left(\frac{\partial C}{\partial \mathbf{C}_\mathrm{bg}}\Delta \mathbf{C}_\mathrm{bg}\right)^2 \tag{22}$$

written in matrix notation as:

$$\mathbf{K}_\mathrm{C} = \Delta \mathbf{C}^2 = \mathbf{w}_\mathrm{elev}\cdot\mathbf{w}_\mathrm{elev}^\top\cdot\mathbf{K}_\mathrm{Celev} + \mathbf{w}_\mathrm{bg}\cdot\mathbf{w}_\mathrm{bg}^\top\cdot\mathbf{K}_\mathrm{Cbg} \tag{23}$$

## 5.2 Emission estimates

Measurements from 18 flights were analyzed to characterize the accuracy of the quantification method. A summary of the estimated emission rates together with the true release rates is presented in Table 2. Estimates are presented for six different quantification methods, which correspond to three different wind treatments applied to two different kriging methods, standard ordinary kriging and cluster-kriging as described above. Among all the methods, the best performing approach, characterized by the lowest RMSE, was obtained by applying Cluster-Kriging Projected-Wind (CKPW), where methane measurements were clustered before kriging, and where the normal components of the instantaneous wind measurements were projected onto the positions of the drone.

A residual plot showing the accuracy of each quantification approach relative to the true release is presented in Fig. 6. The plot illustrates the amount by which we underestimated (negative numbers) or overestimated (positive numbers) the known release for each measurement flight.





**Table 2.** Summary of emission estimates

| Date | Flight Code | Rel. rate [gs$^{-1}$] | Cluster Kriging | | | Ordinary Kriging | | |
|---|---|---|---|---|---|---|---|---|
| | | | Proj. wind | Sca. wind | Log. wind | Proj. wind | Sca. wind | Log. wind |
| 23-Feb | 223_01* | 0.48 ± 0.04 | 0.64 ± 0.56 | 0.66 ± 0.41 | 0.56 ± 0.40 | 1.29 ± 0.98 | 1.26 ± 0.69 | 0.85 ± 0.70 |
| 24-Feb | 224_01 | 0.29 ± 0.03 | 0.79 ± 0.66 | 0.76 ± 0.49 | 0.82 ± 0.51 | 0.60 ± 0.48 | 0.61 ± 0.30 | 0.61 ± 0.31 |
| 25-Feb | 225_01* | 0.29 ± 0.03 | 0.28 ± 0.48 | 0.31 ± 0.41 | 0.29 ± 0.42 | 0.30 ± 0.24 | 0.30 ± 0.11 | 0.29 ± 0.11 |
| | 225_02* | 0.29 ± 0.03 | 0.41 ± 0.46 | 0.45 ± 0.34 | 0.42 ± 0.36 | 0.44 ± 0.33 | 0.45 ± 0.14 | 0.44 ± 0.15 |
| | 225_03* | 0.29 ± 0.03 | 0.30 ± 0.50 | 0.32 ± 0.46 | 0.38 ± 0.56 | 0.48 ± 0.39 | 0.54 ± 0.18 | 0.65 ± 0.18 |
| 08-Mar | 308_02 | 0.26 ± 0.02 | 0.22 ± 0.31 | 0.23 ± 0.30 | 0.27 ± 0.35 | 0.33 ± 0.27 | 0.33 ± 0.22 | 0.40 ± 0.23 |
| 09-Mar | 309_01 | 0.29 ± 0.03 | 0.62 ± 0.37 | 0.77 ± 0.72 | 0.76 ± 0.74 | 0.76 ± 0.33 | 0.89 ± 0.76 | 0.97 ± 0.79 |
| | 309_02 | 0.29 ± 0.03 | 0.39 ± 0.28 | 0.44 ± 0.39 | 0.49 ± 0.42 | 0.42 ± 0.37 | 0.47 ± 0.30 | 0.51 ± 0.31 |
| 12-Mar | 312_01* | 0.31 ± 0.03 | 0.32 ± 0.34 | 0.31 ± 0.28 | 0.32 ± 0.31 | 0.31 ± 0.20 | 0.30 ± 0.09 | 0.32 ± 0.10 |
| | 312_03* | 0.39 ± 0.03 | 0.32 ± 0.53 | 0.32 ± 0.49 | 0.33 ± 0.54 | 0.39 ± 0.26 | 0.39 ± 0.11 | 0.42 ± 0.12 |
| 13-Mar | 313_01* | 0.28 ± 0.02 | 0.15 ± 0.20 | 0.13 ± 0.19 | 0.13 ± 0.18 | 0.22 ± 0.15 | 0.20 ± 0.09 | 0.20 ± 0.09 |
| | 313_02* | 0.41 ± 0.04 | 0.74 ± 0.63 | 0.80 ± 0.60 | 0.91 ± 0.66 | 0.84 ± 0.60 | 0.83 ± 0.36 | 0.97 ± 0.37 |
| | 313_03 | 0.47 ± 0.04 | 0.08 ± 0.16 | 0.09 ± 0.18 | 0.09 ± 0.18 | 0.06 ± 0.06 | 0.08 ± 0.08 | 0.07 ± 0.08 |
| | 313_04 | 0.48 ± 0.04 | 0.13 ± 0.12 | 0.14 ± 0.08 | 0.13 ± 0.09 | 0.14 ± 0.11 | 0.14 ± 0.08 | 0.13 ± 0.08 |
| | 313_05 | 0.52 ± 0.05 | 0.24 ± 0.34 | 0.24 ± 0.32 | 0.20 ± 0.28 | 0.28 ± 0.21 | 0.29 ± 0.15 | 0.25 ± 0.16 |
| 14-Mar | 314_01 | 0.26 ± 0.03 | 0.09 ± 0.08 | 0.09 ± 0.08 | 0.18 ± 0.12 | 0.10 ± 0.07 | 0.10 ± 0.07 | 0.17 ± 0.09 |
| | 314_02 | 0.45 ± 0.05 | 0.02 ± 0.03 | 0.03 ± 0.03 | 0.04 ± 0.02 | 0.05 ± 0.02 | 0.02 ± 0.03 | 0.02 ± 0.04 |
| | 314_03 | 0.68 ± 0.03 | 0.40 ± 0.43 | 0.46 ± 0.48 | 0.26 ± 0.51 | 0.40 ± 0.35 | 0.47 ± 0.36 | 0.24 ± 0.45 |
| NMAE** [%] | | | 53.86 | 57.16 | 58.20 | 64.59 | 68.29 | 71.48 |
| Bias [%] | | | −1.06 | 3.68 | 5.63 | 17.56 | 21.69 | 23.27 |
| RMSE [%] | | | 68.60 | 73.07 | 75.71 | 81.14 | 86.48 | 89.35 |
| Optimal measurement condition | NMAE [%] | | 28.56 | 30.41 | 29.63 | 53.68 | 55.26 | 51.73 |
| | Bias [%] | | 11.44 | 12.05 | 11.90 | 48.34 | 47.53 | 44.35 |
| | RMSE [%] | | 38.66 | 38.40 | 37.88 | 79.55 | 77.95 | 70.48 |
| Non optimal measurement conditions | NMAE [%] | | 74.11 | 78.55 | 81.05 | 73.31 | 78.70 | 87.27 |
| | Bias [%] | | −11.06 | −3.02 | 0.61 | −7.06 | 1.02 | 6.41 |
| | RMSE [%] | | 85.29 | 91.83 | 95.76 | 82.40 | 92.74 | 101.97 |

Flight codes with * are measurement flights with optimal conditions in terms of meteorology and downwind distance. **NMAE: Normalize mean absolute error.

In general, a good agreement between computed estimates using the CKPW approach and true releases was observed as the uncertainty range managed to capture the known release for most measurement flights. A slight overestimation was observed





for most of the earlier flights, but release rates were captured well within the uncertainty range provided by the CKPW ap-
proach. We have observed a systemic underestimation for the last six flights on 13 and 14 March where we did not manage to
capture the true release for four flights (i.e., 313_03, 313_04, 314_01, and 314_02). In order to investigate the reasons for this
underestimation, we compared the predicted kriging fields with a theoretical Gaussian plume dispersion model (see. Fig. S2
and S3) to test whether the vertical and horizontal distance flown by the drone was sufficient to capture the whole plume.
The Gaussian plume model using a Pasquill-Gifford stability class dispersion parameterization scheme provides an analytical
solution for the horizontal and vertical width as a function of downwind distance depending on wind speed and atmospheric
stability. The comparison with the size of the theoretical Gaussian plume suggests that although we managed to detect methane
elevations, we were most likely not able to capture the whole extent of the plume during these flights. The reason is that some
of these flights were conducted at a rather large distance from the source and under low wind conditions, during which the
plume spreads more quickly with downwind distance. For flights 313_03–05, for example, the horizontal and vertical width of
the Gaussian plume computed for the meteorological conditions and downwind distance of the flight were on average $75\,\mathrm{m}$ and
$20\,\mathrm{m}$, respectively. However, the typical cross-sectional plane covered by the drone was of the order of $100\,\mathrm{m} \times 12\,\mathrm{m}$, which
is insufficient to fully capture a spread of the calculated plume, especially with respect to the vertical extent.

The average horizontal and vertical spread of the plume with respect to wind speed and downwind distance computed with
the Gaussian plume model is illustrated in Fig. 7. The spread does not vary smoothly with wind speed, but shows step-wise
changes because the model uses different (but fixed) dispersion parameters for different wind speed and stability classes.
Overlaid on top are dots colored from white to red representing the performance of each measurement flight with lighter
colors showing smaller relative errors. It can be seen that flights with the highest accuracy are the ones that fall within the
blueish region characterized by wind speeds greater than $2\,\mathrm{m\,s^{-1}}$ and a sampling downwind distance ranging from 10 to $75\,\mathrm{m}$.
Measurement flights within this region had a higher accuracy mainly because the vertical spread of the plume was below $10\,\mathrm{m}$,
which is a realistic range for the drone to completely map the plume. For optimal measurement conditions, we found a slight
positive bias of $11\,\%$ using the CKPW method and an RMSE of $39\,\%$. Measurements under sub-optimal conditions had a
smaller average bias (about $-11\,\%$), but a much larger spread with a significant overestimation and underestimation with an
RMSE of $85\,\%$.

All measurement flights were also analyzed using an ordinary-kriging (OK) algorithm, where methane measurements were
not clustered before kriging. By doing so, each measurement flight was fed directly into an GMM to determine the hyper-
parameters for kriging. Likewise, Matèrn 5/2 covariance kernel was used to quantify the correlation between the measured
data. Ordinary kriging produces a single methane field with expected value and variance because a single correlation length
scale is assumed for both the background and the plume data. The assumption of a single correlation length leads to a strong
smoothing of the plume (Stachniss et al., 2009), as illustrated in Fig. 5d. Obtained methane fields were combined with the
same three different wind treatments to compute the release rates. A summary of emission rates computed using ordinary
kriging is presented in Table 2, and the range of the residuals for each quantification approach is illustrated in Fig. 8. It shows
that cluster-based kriging, in general, outperforms ordinary kriging as evidenced by lower RMSE and lower relative absolute
errors. On average, all data treatments tend to overestimate the true release, but the lowest overestimation was obtained using





the CKPW approach. Generally, a larger variability of residuals (wider inter-quartile band) was obtained for the approaches

using OK as compared to the respective CK counterpart. A concrete example to see the difference between the reconstructed methane plume using cluster kriging and ordinary kriging is presented for flight 312_03 in Fig. 5c and Fig. 5d. CK proves to better preserve the shape of the plumes, which results in a better accuracy of the estimates.

### 5.2.1  Impact of altitude uncertainties on emission estimates

Initially, the altitude measurements of the drone-based system were relying exclusively on the on-board internal GPS, but later

it became evident that this has some impact on our capability of emission estimates. The RTK-GPS system was implemented a few days after the start of the MATRIX campaign, and 11 out of 18 measurement flights contain both UAV altitude and RTK altitude. We observed an average drift of the UAV-GPS of $0.10\,\mathrm{cm\,s^{-1}}$ which translates to an altitude error of about $0.6\,\mathrm{m}$ for a 10-minute flight duration. This drift is consistent with the uncertainty reported by the drone manufacturer, though sometimes error were larger of up to $0.20\,\mathrm{cm\,s^{-1}}$ (see Table S1). An erroneous altitude retrieval on certain flight levels may

lead to a distortion of the emission plume, which ultimately affects the estimated emissions (see Fig. S4). A summary of the percentage difference between the emission estimates derived using two different altitudes is presented in Fig. 9. Differences are in the range of $-8$ to $18\,\%$ with an absolute average difference of $4\,\%$, suggesting that the errors introduced by inaccurate vertical positioning are relatively small compared to the overall uncertainty of the CKPW quantification method. The highest differences occurred on flights 313_02 and 313_05, during which the drift of the UAV-GPS was particularly large (about

$0.17\,\mathrm{cm\,s^{-1}}$, see Table S1). These findings are important aspects also in the context of the ROMEO campaign, during which the high-accuracy RTK-GPS system was not yet implemented. Now, it can be stated that the emissions reported for the ROMEO campaign should have a similar accuracy as presented here, at least for those cases, where meteorological conditions were favorable.

### 5.2.2  Impact of wind speed and direction on emission estimates

Similar to our study, Yang et al. (2018) performed a rasterized mass-balance approach to quantify emissions from individual gas wells in Texas, USA using UAVs. Based on their results, they proposed a minimum threshold of wind-speed of $2.3\,\mathrm{m\,s^{-1}}$ and wind direction variability not greater than $33.1°$ in order to quantify emissions with an accuracy of better than $50\,\%$. Applying the same threshold criteria and additionally restricting the measurements to a maximum downwind distance of $75\,\mathrm{m}$, we have identified 8 out of 18 flights from our campaign that satisfy these criteria (see Fig. S5). As illustrated in Fig. 8,

these flights indeed exhibit a lower RMSE and absolute mean error. RMSE and absolute error were reduced to $39\,\%$ and $29\,\%$ respectively as compared to $69\,\%$ RMSE and $54\,\%$ absolute error for all flights. Computed emission rates were on average slightly overestimated by $11\,\%$. In contrast, a lower average accuracy was observed when measurement flights were performed under less favorable wind conditions. Computed emission rates under these conditions were generally underestimated by $11\,\%$ with a higher corresponding RMSE and absolute mean error of $85\,\%$ and $74\,\%$. Underestimation of true releases during highly

variable weather conditions may be attributed to incomplete sampling of methane plumes as discussed above. Variability of



residuals (width of inter-quartile band) among all approaches is significantly lower for measurement flights under optimal conditions as compared to measurements performed in sub-optimal conditions.

## 5.3 Comparison of AirCore and QCLAS emission estimates

Having simultaneous samples of methane plumes using the QCLAS and AirCore systems, we have found that the AirCore measurements were smoothed by an average of $20\,s$ $(1\sigma)$ using a Gaussian smoothing function when compared with measurements using the QCLAS. We also observed that AirCore measurements are temporally shifted by an average of $7\,s$ and stretches linearly with time at an average rate of $0.06\,s$ for every second of QCLAS measurement. The smoothing, stretching, and shifting parameters obtained for each individual flights are presented in Table 3. Corrected and original AirCore methane measurement flights were subjected to CKPW quantification approach to compare how the stretched and shifted AirCore measurements affect the quantifications. Emissions are compared to emission estimates using QCLAS measurements to see the degree of agreement between the two systems. A summary comparing the differences in emission estimates is presented in Table 3. We have observed that the emission estimate computed using the corrected time-series is $3\,\%$ more accurate compared to its original counterpart. Nevertheless, the uncertainty bounds of most quantification flights manage to capture the true release. In extreme cases, where the time shift and stretching is not sufficiently well known, the size and location of the plume might not be captured accurately. As an example, a comparison of reconstructed plume with and without applying proper correction for flight 312_03 is illustrated in Fig. S6. The figure shows that the uncorrected reconstructed plume tends to be cut on the left side of the mapping plane. After applying the proper correction, the plume shifted to the right, putting the methane plume closer to the center of the mapping plane. This resulted in a $23\,\%$ increase in emission estimate, bringing it much closer to the actual release. Thus, even though uncertainty bounds manage to capture most of the releases, accounting for the proper time shift and stretching of the AirCore data is important when performing a mass-balance quantification approach, especially in extreme cases.

## 5.4 Comparison with other methodologies

A direct comparison with another method was performed for the OTM-33A method. Quantified releases using OTM-33A and our mass-balance approach are summarized in Table 4. Although the number of simultaneous quantifications is limited, the results show that both approaches are close to the true-release and that the uncertainty bounds of both methods usually capture the true-release. This showcases that our drone-based quantification technique has a great potential and is at par in measuring $CH_4$ emissions from oil and gas wells when compared with the OTM-33A method. Emission estimates using OTM-33A for flight 225_01–03 were identical because OTM-33A estimates are more robust if the input data lasts longer than 20 minutes. Since the release rate during that day was constant and continuous, one emission estimate was used for the three drone-flight emission estimate for that day.

Table 5 compares the uncertainty of our UAV-based quantification method with other methods as previously summarized by Caulton et al. (2018). With an accuracy ranging from $28\,\%$ to $75\,\%$, our method is at par with existing quantification techniques, specifically with mass-balance approaches using aircrafts/UAVs. A major advantage of our UAV-based method is that it can





**Table 3.** Correction parameters and calculated emission rates for AirCore measurements

| Flight | Correction Parameters | | | Release | CKPW Estimates | | |
| Code | Shift | Stretch | Smoooth | Rates | QCLAS | Corr. AirCore | Orig. AirCore |
| | [s] | [AirCore(s)/QCLAS(s)] | [s] | [gs$^{-1}$] | [gs$^{-1}$] | [gs$^{-1}$] | [gs$^{-1}$] |
|---|---|---|---|---|---|---|---|
| 312_01 | 1.79 | 0.03 | 20.77 | $0.31 \pm 0.03$ | $0.32 \pm 0.34$ | $0.31 \pm 0.40$ | $0.30 \pm 0.42$ |
| 312_03 | 6.32 | 0.04 | 27.29 | $0.39 \pm 0.03$ | $0.32 \pm 0.53$ | $0.25 \pm 0.26$ | $0.20 \pm 0.60$ |
| 313_02 | 10.27 | 0.10 | 19.03 | $0.41 \pm 0.04$ | $0.74 \pm 0.63$ | $0.58 \pm 0.85$ | $0.65 \pm 0.92$ |
| 313_04 | 7.22 | 0.05 | 19.61 | $0.48 \pm 0.04$ | $0.13 \pm 0.12$ | $0.15 \pm 0.25$ | $0.17 \pm 0.31$ |
| 314_01 | 12.81 | 0.06 | 17.90 | $0.26 \pm 0.03$ | $0.09 \pm 0.08$ | $0.12 \pm 0.18$ | $0.13 \pm 0.20$ |
| 314_02 | 2.01 | 0.05 | 18.11 | $0.45 \pm 0.05$ | $0.02 \pm 0.03$ | $0.02 \pm 0.04$ | $0.04 \pm 0.04$ |
| | $6.73 \pm 4.41$ | $0.06 \pm 0.02$ | $20.45 \pm 3.51$ | NMAE [%] | 55.92 | 49.75 | 52.45 |
| | | | | Bias [%] | $-28.02$ | $-35.34$ | $-32.77$ |
| | | | | RMSE [%] | 65.24 | 57.54 | 58.61 |

**Table 4.** Emission rates from QCL-CKPW and OTM-33A in g s$^{-1}$

| Fl. Code | Release | CKPW | OTM-33A |
|---|---|---|---|
| 223_01 | $0.48 \pm 0.04$ | $0.64 \pm 0.56$ | $0.53 \pm 0.17$ |
| 224_01 | $0.29 \pm 0.02$ | $0.79 \pm 0.66$ | $0.26 \pm 0.09$ |
| 225_01 | $0.29 \pm 0.03$ | $0.28 \pm 0.48$ | $0.47 \pm 0.17$ |
| 225_02 | $0.29 \pm 0.03$ | $0.41 \pm 0.46$ | $0.47 \pm 0.17$ |
| 225_03 | $0.29 \pm 0.03$ | $0.30 \pm 0.50$ | $0.47 \pm 0.17$ |

be applied to sources that are not easily accessible and where no road is present in a suitable distance perpendicular to wind
direction for ground-based mobile measurements. Another advantage is that it can be applied to quantify the total emissions of
a cluster of sources, provided that the UAV can map the full extent of all individual source plumes. Ideally, the emission from
an individual source should be quantified multiple times. The individual estimates provide an invaluable measure of uncertainty
in addition to the method uncertainty estimated here for individual flights. This is even more important under highly unstable
and turbulent conditions, since an individual flight can only capture a snapshot of a turbulent plume.

**6   Conclusions**

A novel strategy of methane flux quantification with the use of unmanned aerial vehicles (UAVs) equipped with a methane
sensor has been developed and applied to an extensive tracer release experiment. Real-time atmospheric methane mole frac-
tions were measured in-situ using a Quantum Cascade Laser Spectrometer (QCLAS) and an Active AirCore system. Both





**Table 5.** Uncertainty of different $CH_4$ emission quantification techniques.

| Approach | Uncertainty estimate | Literature |
|---|---|---|
| Ground-based thermal imaging | 3–15 % | Gålfalk et al. (2016) |
| Chamber sampling | 5–60 % | Allen et al. (2013, 2015); Kang et al. (2014); Yver Kwok et al. (2015) |
| Tracer ratio technique | 20–50 % | Lamb et al. (2015, 2016); Roscioli et al. (2015) |
| | | Subramanian et al. (2015); Zimmerle et al. (2015); Omara et al. (2016) |
| | | Feitz et al. (2018); Fjelsted et al. (2020) |
| Airborne mass-balance | 20–75 % | Karion et al. (2013, 2015); Nathan et al. (2015); Caulton et al. (2018) |
| | | Shah et al. (2020) |
| *Airborne CKPW mass-balance* | 30–77 % | |
| | | Golston et al. (2018); Yang et al. (2018); Shah et al. (2020) |
| Ground-based stationary dispersion | 25–66 % | Brantley et al. (2014); Robertson et al. (2017); Edie et al. (2020) |
| Ground-based mobile dispersion | 50–350 % | Ars et al. (2017); Weller et al. (2018) |
| | | Bakkaloglu et al. (2021); Defratyka et al. (2021) |

instruments are lightweight and have a compact footprint, allowing them to be mounted on commercially available drones.

Emissions were quantified by applying a cross-sectional mass-balance approach. An extensive tracer release experiment was conducted in Dübendorf, Switzerland from 23 February to 14 March 2020 to develop, optimize, and evaluate the method. In addition, source quantification from the drone were compared for selected cases with results from stationary measurements applying the OTM-33A method.

The mass-balance approach was performed by flying the drone-integrated system at a cross-section downwind of the source

at multiple vertical levels. Methane mole fraction measurements were subject to two different data-treatments, while the wind measurements were treated in three different ways, thus giving us in total six methane-quantification approaches. Each of these were applied to all flights and evaluated for their ability to reproduce the true releases.

During the campaign, 18 flights suitable for emission quantification could be performed. Among the six quantification approaches, the best results were obtained by using the CKPW (cluster-kriging with projected wind) approach. The true release

could be estimated with a normalized mean absolute percentage error of 54 %. The highest absolute percentage error of 71 % was obtained using the OKLW (ordinary-kriging with logarithmic wind profile) approach. A consistent underestimation of methane fluxes occurred in our quantification approach when the mass-balance method was performed at a downwind distance of more than 75 m. Simulations with a simple Gaussian plume model suggest that we were most likely not able to capture the whole extent of the plume during these flights, especially with respect to its vertical extent. Comparison of QCLAS-CKPW

emission estimates with quantified emission rates using an independent ground-based quantification technique, OTM-33A, shows that both methods captured the true release almost every time.

A general guideline, when performing drone-based emission quantification of emission sources, requires favorable wind conditions with a minimum wind speed of $2.3\,\mathrm{m\,s^{-1}}$ and a maximum wind direction variability of 33.1°. Under these condi-





tions, measuring at a downwind distance of $75\,\mathrm{m}$ ensures the true emission can be fully mapped both horizontally and vertically.
In cases where an RTK-GPS is not present, a vertical spacing of at least $0.5\,\mathrm{m}$ is recommended to properly account for the average drift of commercial UAV-GPS of about $0.11\,\mathrm{cm\,s^{-1}}$.

Having a high-precision and fast $CH_4$ analyzer, such as the QCLAS, offers the benefit of correctly mapping the methane plume both spatially and temporally as compared to other methods such as collecting air samples with subsequent analysis on the ground. In extreme cases, poor mapping of the emission may ultimately lead to over- or underestimation of its value.
This is evidenced in one of the measurement flights, i.e., 312_03, where a reconstructed methane plume using the uncorrected AirCore measurement resulted in a significant underestimation (about $48\,\%$) of the true-release. Nevertheless, the uncertainty bounds of the CKPW quantification approach usually manage to capture the true release.

In conclusion, drone-based emission quantification using the CKPW approach proved its capability to quantify emission fluxes from methane point-sources. This approach can be easily scaled-up to confidently quantify total emissions for a cluster
of sources given that the drone-system can map the full extent of all individual plumes. The use of UAVs in quantifying localized methane sources offers an advantage of allowing additional freedom of sampling locations where stationary monitors and ground-based mobile sensors cannot be deployed. It also allows rapid adjustment to changing wind conditions, which proved to be particularly beneficial during the ROMEO measurement campaign, where a large number of oil- and gas wells had to be quantified in a short amount of time.

*Code availability.* The cluster-based kriging package used to process our drone measurements is written in Python 3.7.4 and is available upon request. The codes will be available on Gitlab after the final revision of the manuscript

*Data availability.* The data is available upon request to Randulph Morales (randulph.morales@empa.ch) for the discussion paper and will be made available on https://zenodo.org for the final revised paper

*Author contributions.* RM implemented the cluster-based kriging approach for drone measurements, applied and validated the approach to
all drone-based measurements and wrote the manuscript with input from all the co-authors. BT and LE developed the in-situ QCLAS that was mounted on the drone and designed the release experiment. KV and HC developed and operated the AirCore system that was mounted on the drone. RM, JR, and KV performed all the measurement flights during MATRIX. PK and MS performed all OTM33A and the subsequent analysis of the ground-based data. SH implemented the RTK-GPS system for the drone. LE, HC, and DB supervised the implementation of MATRIX. DB, BT, LE, HC, and MS provided critical feedback to the study and reviewed the manuscript. DB supervised the whole study.

*Competing interests.* The authors declare that they have no conflict of interest





*Acknowledgements.* This work was supported by the ITN project Methane goes Mobile – Measurements and Modelling (MEMO2; https://h2020-memo2.eu/), funded by the European Union's Horizon 2020 research and innovation programme under the Marie Sklodowska-Curie grant agreement No 722479.



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





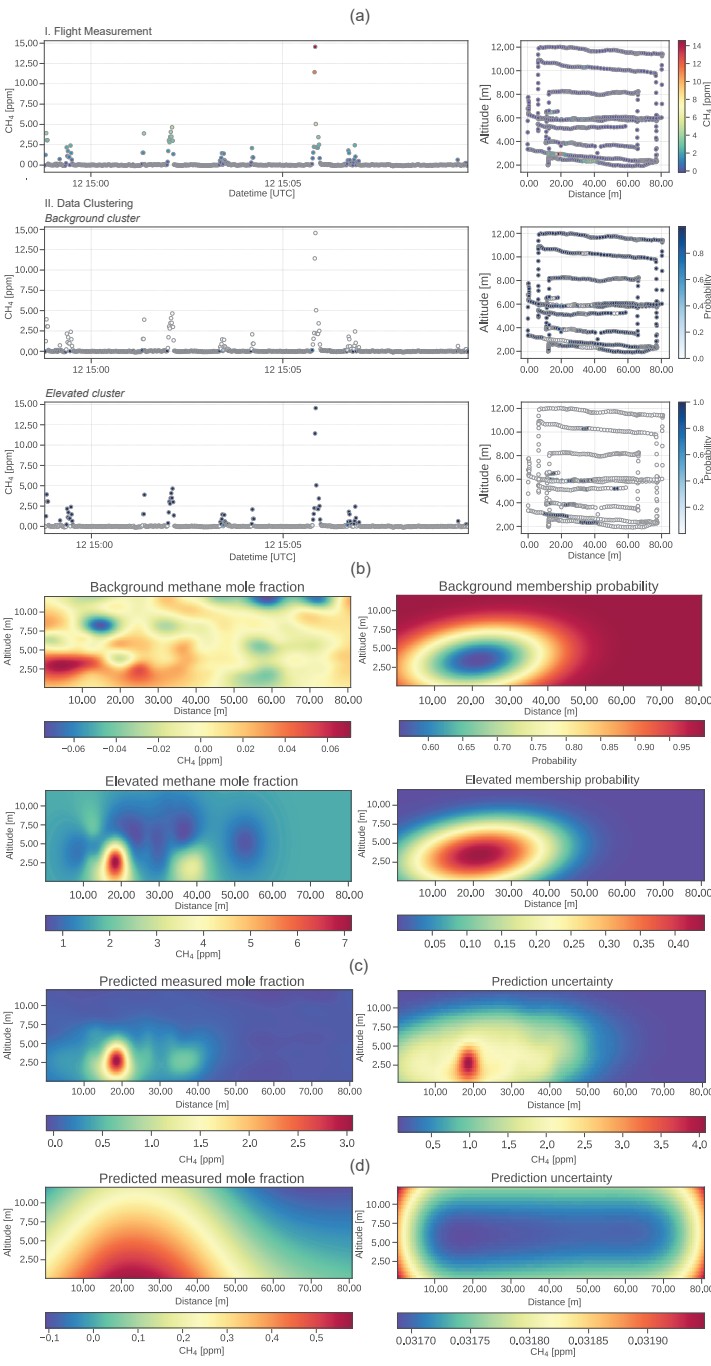

**Figure 5.** (a) Clustering result for flight 312_03 on 12 Mar 2020 after applying GMM with two mixture components. The background and elevated cluster complement each other; the total probability of each data-point shared between the two clusters is equal to one. (b) Kriging prediction and membership probabilities of each spatial point within the domain of interest for background and elevated clusters. (c) Expected value and variance of methane mole fractions after combining kriging prediction of the two clusters and their respective membership probabilities. (d) Expected value and variance of methane mole fractions using ordinary kriging.





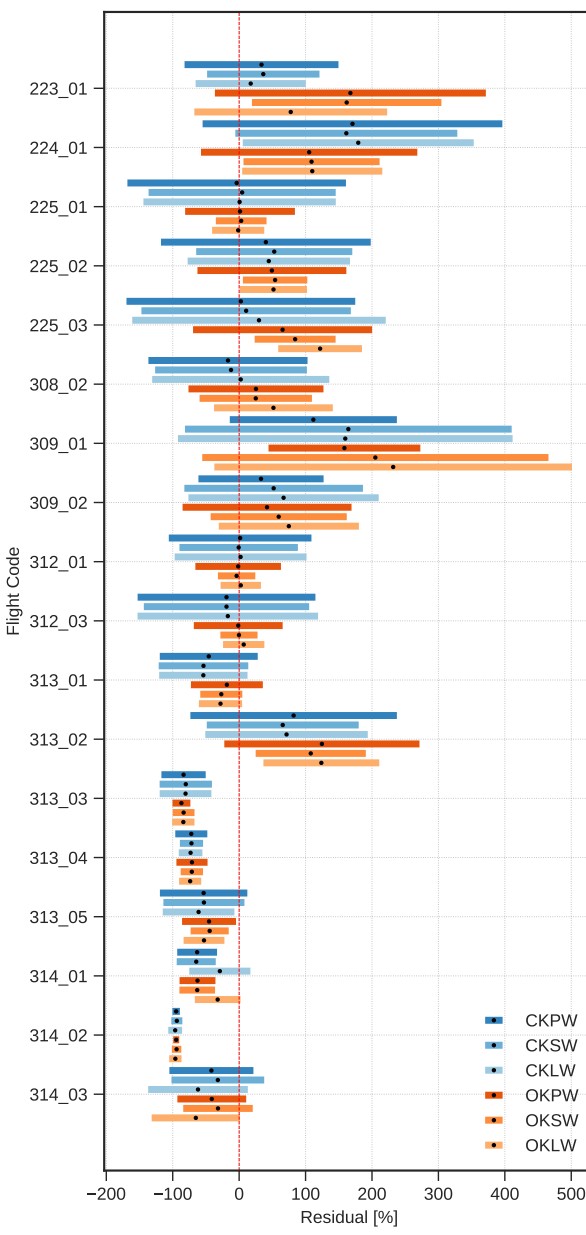

**Figure 6.** Residual plot. Color-coded solid bars represent the range of residuals using different quantification approach with the mean value represented as black dots. Values to the right of the red line correspond to overestimations, values to the left correspond to underestimations. CK and OK stands for Cluster-Kriging and Ordinary-Kriging, respectively. PW (projected wind), SW (scalar wind), and LW (logarithmic wind) refer to the different wind data treatments.





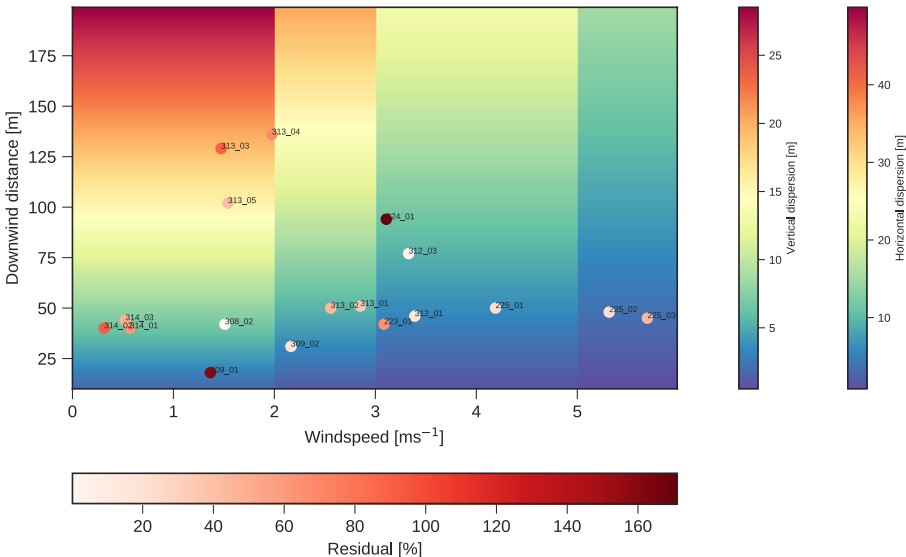

**Figure 7.** Theoretical horizontal and vertical spread of a plume with respect to wind speed and downwind distance. White to red dots refers to the individual error of each quantification flight, lighter being more accurate than darker dots.

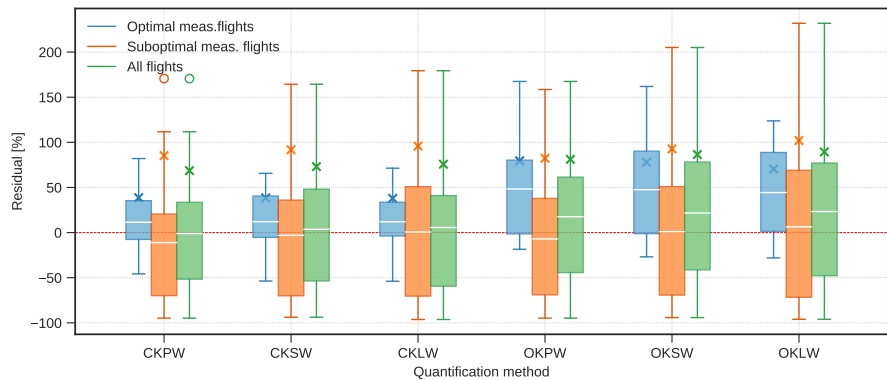

**Figure 8.** Color coded box-plots represent the range of residuals of measurement flights grouped according to meteorological and threshold conditions. Solid white lines represent the mean bias and the $\times$ mark represent the RMSE for each quantification approach.





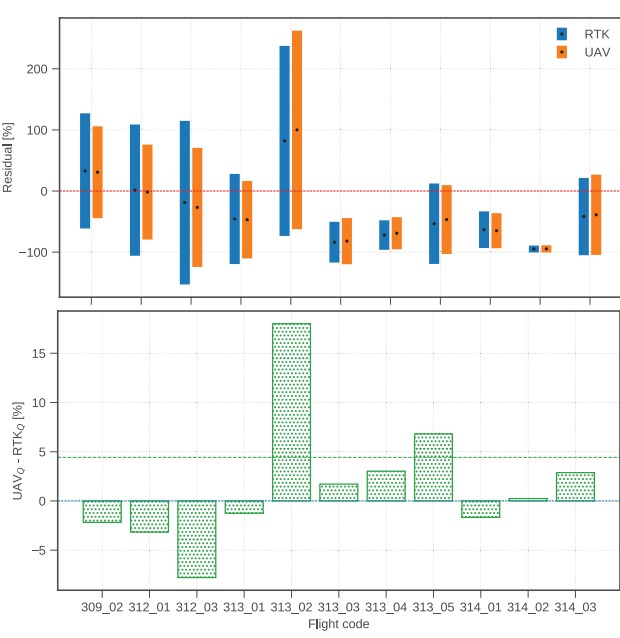

**Figure 9.** Difference in emission estimates using two different GPS altitudes. The green dashed line represents the absolute average difference between the two estimates.