# Peer review of "Controlled release experiment to investigate uncertainties in UAV-based emission quantification for methane point sources"

_Atmospheric Measurement Techniques, 2021_

## Referee Comment (RC1)

**Review of amt-2021-314**

"A tracer release experiment to investigate uncertainties in drone-based emission quantification for methane point sources" by Morales et al.

This paper presents a set of experiments using controlled methane releases to validate the mass balance approach to flux quantification when applied to UAV-based sampling of methane. The work rigorously characterises sources of uncertainty, bias, and error with respect to interpolation methods, instrumentation, and treatment of atmospheric transport. As such, this represents a useful contribution to UAV-based flux quantification methodology, and provides valuable guidance to others attempting similar emission measurements. However, I have a few major concerns regarding the validity of the methane measurements presented. The work appears to present methane mole fractions below the global atmospheric background mole fraction (of roughly 1.9 ppm; see e.g. Lan et al., Phil. Trans., 2021). Furthermore, calibration of the three instruments used to measure methane is not mentioned. I support publication of this work once these, and the additional comments and suggestions outlined below, have been implemented.

*General comments*

- I recommend swapping the phrase 'tracer release' for something more appropriate such as 'controlled release', especially in the title. So-called tracer techniques rely on the concurrent measurement of the target gas (in this case methane), and a tracer gas, and are generally referred to as tracer ratio, tracer dispersion or tracer release methods in the literature (see e.g. Mønster et al., Waste Management, 2014; Yacovitch et al., Elem. Sci. Anth., 2017 and various references in the authors' own introduction). Such an experiment, involving a tracer gas, was not performed here. Validation of plume mapping and flux methodology using a source with a known emission rate (as done in this work) is usually referred to as a controlled release experiment (see e.g. Thorpe et al., Remote Sensing of the Environment, 2016; Heltzel et al., Environments, 2020; Shah et al., Atmos. Meas. Tech., 2020).

- Drone and UAV are used interchangeably throughout the text. It should be made more clear, somewhere early on, that the two refer to the same thing and one term (probably UAV) used consistently.

- Figure text is often very difficult to read without considerably zooming in. All figure text needs to be much larger.

- Calibration of the three instruments, with respect to measurements of methane, is not mentioned at all in the text. Was calibration to a methane standard (World Meteorological Organization) performed at all, and if so what was the calibration procedure? If no calibration was performed, this throws the validity of the results into question.

- The OTM-33A method usually uses instrumented vehicles and mobile sampling to quantify a flux, although examples do exist in the literature of stationary measurements (e.g. Foster-Wittig et al., Atmospheric Environment, 2015; Shaw et al., JAPCA Waste Management, 2020). Could the authors' comment on the possibility of applying the mobile OTM-33A method to the UAV-based measurements (has this possibility been explored)?

- The plots appear to show $CH_4$ mole fractions of 0 ppm (Figure 3 and 5). Unless these plots are actually showing $\Delta CH_4$ ($CH_4$ − background), this is impossible. The tropospheric

background mole fraction of methane is roughly 1.9 ppm (e.g. Lan et al., Phil. Trans., 2021). Could the authors explain these results?

- Abbreviations: Abbreviations for the six methods (CK, OK, PW, LW and combinations thereof) are used inconsistently. It would be useful to the reader for them to be introduced more distinctly in the methods section and then used consistently throughout the results and in figures/tables.

- Did the authors consider measuring wind speed and wind direction in situ from the drone by attaching an anemometer? Concerning wind speed estimation/interpolation, would there be any improvement using a combination of methods two and three (LW and PW)? Further, in Section 4.3, I would recommend definitively stating the abbreviated definitions used to refer to the three approaches later in the text (for example "Proj. wind" Table 2, or "PW" later).

- How representative is the range of controlled release rates used here ($0.2 - 0.7$ g s$^{-1}$) of true emissions from oil and gas facilities, or other methane sources? I would expect real emissions to have a much greater range, and that the release rates used are at the lower end of that range. Are the authors' conclusions (for example, on wind speed, wind direction, and distance from plume limits) therefore only applicable to the controlled release rates used in this work, or are they equally applicable to emissions tens, or hundreds, of times stronger? If this is not the case, then the conclusions should be caveated by stating that these results are for a limited range of emission strengths.

*Specific comments*

- L13: It is not clear to me what "stretched by 7 s and 0.06 seconds for every second of QCLAS measurement, respectively" is referring to here. This phenomenon is better explained in Section 5.3 and the authors should consider amending the abstract text to avoid confusion.

- L105: For comparison, it would be useful to include the instrument measurement precision for the Picarro CRDS (as mentioned for the QCLAS system on L83).

- L129: Could "not too strong winds" be quantified here e.g. greater than X m s$^{-1}$? I also assume this was due to the limitations of the UAV system used? Explaining the reason behind this limitation would be useful for guiding others.

- Figure 3: The wind rose is exceptionally small and doesn't add much information to the figure in its current form. The wind rose might be better viewed in a separate panel, adjacent to the top-down view of the CH$_4$ data. The wind rose is also not mentioned in the figure caption.

- Figure 3: Is the orange line showing the source-transect distance? This is not clear and should be made clear in the figure caption.

- Figure 3: The figure may benefit from an additional arrow illustrating the average wind direction for this flight (which ties in with the wind rose).

- L195: As for the Picarro instrument, it would be useful to include instrument characterisation (measurement precision etc.) here for the LI-COR instrument.

- L219: Extra "to" in "matching the timestamp of the anemometer to the to GPS location".

- L230: For clarification, were background $CH_4$ mole fractions measured upwind of the emission source, or from either side of the emission plume? It might be useful to present the measured background mole fraction value(s) and uncertainty somewhere.

- L237: Missing the word "to" between "due" and "the".

- L271: Two identical references on this line – Tadić et al., 2015.

- L285: Missing the word "one" between "only" and "cluster".

- Figure 5: Could the caption include which instrument was used for the methane measurements shown?

- Figure 5: The amount of panels here makes readability particularly difficult. I would recommend splitting into two separate figures: Fig. 5a as a single figure, and Fig. 5b, 5c, and 5d as a single figure. It may also be useful to have a direct side-by-side comparison of actual in situ measured $CH_4$ (showing sparse spatial distribution on the vertical plane) alongside the 'predicted measured' Krigged $CH_4$.

- Fig. 5b: As in general comments above, here $CH_4$ mole fraction is in a range of -0.06 to +0.06 ppm. These values are impossible (especially the negative values) in the troposphere. Could the authors explain these results?

- I would recommend moving equations 17 through 23 (and surrounding text) to a relevant section(s) in the methods section, as this is more Methodology than Results.

- Table 2: This table is difficult to read due to the sheer amount of values. The information is much better visualised in a plot such as Figure 6. I would consider moving the full table to the Supplement, and only including the overall results (NMAE, Bias, RMSE) for all six methods in the main manuscript.

- Table 2: Abbreviations for the six methods (e.g. CKPW) are used throughout the text but not in this table.

- Figure 8: The caption should probably mention that these are residuals in flux estimates.

- Section 5.2.2: Could this section refer to Figure 8 as well, and the comparison of different meteorological regimes?

- Table 4: Would it be useful to present the NMAE, bias, and RMSE in this table, as done in comparisons of the AirCore with the QCLAS results (Table 3), and for the comparison of the six drone-based methods (Table 2)?

- The link in the reference for US EPA 2014 goes to a page which states that "Emissions Measurement Center has Moved" – the link might need to be corrected.

- Figure S6: Would it be worth showing the plume constructed from the QCLAS data too, for comparison?

Lan et al., Phil. Trans., 2021 https://doi.org/10.1098/rsta.2020.0440
Mønster et al., Waste Management, 2014 http://dx.doi.org/10.1016/j.wasman.2014.03.025
Yacovitch et al., Elem. Sci. Anth., 2017 https://doi.org/10.1525/elementa.251

Thorpe et al., Remote Sensing of the Environment, 2016 https://doi.org/10.1016/j.rse.2016.03.032
Heltzel et al., Environments, 2020 https://doi.org/10.3390/environments7090065
Shah et al., Atmos. Meas. Tech., 2020 https://doi.org/10.5194/amt-13-1467-2020
Foster-Wittig et al., Atmospheric Environment, 2015
http://dx.doi.org/10.1016/j.atmosenv.2015.05.042
Shaw et al., JAPCA Waste Management, 2020 http://dx.doi.org/10.1080/10962247.2020.1811800

---

## Referee Comment (RC2)

Review of "A tracer release experiment to investigate uncertainties in drone-based emission quantification for methane point sources"

Summary
The manuscript presents results from controlled release experiments, with methane concentrations and subsequently emissions calculated with a drone-based methodology. A DJI Matrice 600 drone is equipped with a low-cost RTK system, in-situ methane analyzer, and an AirCore for post-flight concentration analysis with a Picarro analyzer. Corrections for the AirCore sampling timing are presented, along with analysis of the drone altitude accuracy. To calculate emission rates, different variations of the mass balance technique are presented with comparison in some cases to OTM33A. In addition to these comparisons, a new technique based on cluster kriging is described.

Overall evaluation
The Introduction/references, scope of work, and scientific approach of the work are good.

Some issues I found already mentioned by Reviewer 2 include finding L13 of the abstract confusing about the stretching by 0.06 seconds, wondering whether the methane data in all the plots are already background-subtracted, and suggesting improvements to the general readability of figures (other than Fig 1).

Regarding the novel cluster kriging approach adapted here, the paper cited by van Stein et al. 2020 concluded the method is designed to 'reduce the time and space complexity of the Kriging method'. While dividing into elevated and background clusters makes sense, I do wonder how the above statement fits in. Specifically, if the difference between cluster kriging and ordinary kriging shown here has less to do with the theoretical basis of the method, and more a difference in the parameters used given that Fig 5(d) on left has a significantly different appearance (length scale or search radius?) than that of Fig 5(c). Please explain. In general, while the math is presented if Section 4.5.2, I think some of the more practical details could be mentioned. Does the cluster kriging python package mentioned in the code availability statement also perform ordinary kriging, or that comes from elsewhere? Maybe add an example of the semivariogram or kriging parameters in the supplement to better illustrate the method?

Some minor details were unclear, especially regarding the altitude analysis and AirCore correction, as detailed below.

Specific comments
L27 Gurney et al. 2021 in *Nature Communications* is likely the wrong reference here. That paper is focused on FFCO2 from cities, not CH4 from oil and gas.

L48-50 Shaw et al. 2021 ("Methods for quantifying methane emissions using unmanned aerial vehicles: a review") and/or Hollenbeck et al. 2021 ("Advanced Leak Detection and Quantification of Methane Emissions Using sUAS") could be also considered adding here, as they are recent reviews on the subject of UAV methane quantification.

L114 'capturing raw streams' – This is perhaps too vague. I think it is not so much a raw stream as a different stream (the carrier phase), see https://novatel.com/an-introduction-to-gnss/chapter-5-resolving-errors/real-time-kinematic-rtk

Figure 2: The 14 on the x-axis tick labels is unneeded (see presumably matplotlib.dates.DateFormatter)

Figure 2: Isn't 0 AGL [m] defined as the takeoff altitude for UAV-GPS by the Matrice? Also, are takeoff and landing locations here different or the same?

Figure 2: The pressure altitude is impressively consistent comparing against RTK altitude. The spikes seen in the bottom panel of Figure (2) could be a little misleading since they appear to be caused simply by small differences in timing relative to the RTK during ascent and descent where altitude is changing quickly.

Figure 2 caption – the meaning of subscript m in bottom panel legend could be mentioned (slope from linear regression). They must also have some impact on the pressure altitude drift estimate unless robust regression was used

L144 Later, the make/model of 3D sonic anemometer is mentioned (uSonic-3 Scientific). What was the type of 2.5D anemometer?

Table 1 stability here is presumably based on equation from L215, not the Pasquill stability classes, which are also mentioned (L180)

L250 It's a little unclear if the 3S algorithm is new to this manuscript, or if it is presented in the two manuscripts cited on L247 that are in preparation / review. With being able to read those, the writing here is a is a little hard to follow. A simple 1D Gaussian smoothing function need only have one parameter – a standard deviation. How does $F(x,b)$ accommodate three parameters?

Figure 4 legend – 'Stretch' is written twice. Is one of them supposed to be shift? Also, the 0.06 s/s stretching is mentioned in abstract, but the other two numbers (12.81 and 17.90) are different?

Figure 4 – Frankly, the algorithm mainly just seems to correct for the shift, also called time lag by some other authors. Are the other two parameters really helpful?

L361 16.04 kg should be g for the molar mass of methane

Table 2 - In footnote about optimal conditions, suggest mentioning they are defined in Section 5.2.2

Table 5 – Suggest putting 'This study' (or similar) in the column next to Airborne CKPW mass-balance. Or somehow clarify, since the studies mentioned here - Golston et al. (2018); Yang et al. (2018); Shah et al. (2020) - do not use CKPW

L529 'Under these conditions, measuring at a downwind distance of 75 m ensures the true emission can be fully mapped both horizontally and vertically'. This is a little confusing, since it sounds like you need to be >= 75 m downwind to fully capture the plume, while L523 indicates underestimation at those distances.

Figure 6, 8, and 9 show 'residuals' in %, which here must mean the percentage error of the estimate versus the known controlled release amount (but without calculating absolute values). Where does the 'range of residuals' come from?

L742 Suggest replacing the dead link to U.S. EPA with the citation given at https://cfpub.epa.gov/si/si_public_record_Report.cfm?Lab=NRMRL&dirEntryId=309632]

---

## Referee Comment (RC3)

This manuscript describes the development and evaluation of a new cluster-based kriging approach for mass balance calculation of facility-level emission rates, using UAV-based measurements. Easily-deployable techniques that enable accurate determination of facility-level emissions have an important role to play in both improving emission accounting (e.g. national emission inventories) and compliance monitoring, as many countries move towards tighter regulation of methane emissions. The results of the controlled-release experiments detailed in this study suggest that this approach could be an important tool in such efforts going forward, especially as lightweight instrumentation for accurate $CH_4$ measurement becomes widely commercially available. Comparisons of emission estimates calculated using different measurement techniques, kriging approaches and assumptions regarding the local wind field are instructive for future studies, as is the comparison against the established OMT33A method. The paper is in general very clearly written, and I suggest that it should be published in AMTD after the following minor points have been addressed.

In general I've tried not to repeat comments already made by the other reviewers, but I do agree with RC1 that a brief discussion of calibration is required. I also think it would help to clarify things if the term "$CH_4$ enhancement" were to be used in cases where background values have been subtracted from the data (which I think is pretty much everywhere). Perhaps a couple of extra sentences briefly summarising the application of the REBS algorithm would be useful too (especially with regard to the RC1 question concerning where the background measurements were taken - I assume the answer is anywhere on the downwind measurement plane that the REBS algorithm identified)?

L27 - Alvarez et al. (2018) would be a more appropriate reference here (Gurney et al., 2021, is definitely wrong), although there are more recent options that would do the job too.

L181 - this reference (U.S. EPA, 2014) needs working into the sentence, which currently doesn't make sense without the bracket.

L202 - the star on the friction velocity should be a subscript (as in Equation 4).

L226 - I'm a bit confused as to why this step was necessary. If both the QCLAS and RTK-GPS received GPS signals, why were they not already synchronised on GPS time?

L230 - in addition to my general point above, it would probably be clearer to say that background $CH_4$ mole fractions were "subtracted" instead of "removed".

Equation 5 - I understand that this approach is based on previously published work, but the application is sufficiently different that it would be useful to provide some more information here. I suggest explicitly stating the form of **F**. Have I got it right that the parameter vector **b** consists of the QCLAS measurements? If so I would also state that explicitly.

Figure 4 - Somewhere in either the caption and/or the associated main text it should be explicitly stated that these values were optimised separately for each flight.

L315 - maybe I missed something, but is it explained anywhere how the data are hard-clustered prior to performing ordinary kriging?

L322 - I have no doubt that the Matèrn covariance kernel is a valid choice here, but as a general comment I feel that the choice of kernel should be based on an examination of the specific dataset on which kriging is being performed (although of course it can be guided by previous studies/experience). I'm sure that such examination was performed (i.e. someone checked to make sure the optimised function was a reasonable fit to the data for each flight) - I'm happy to leave it up to the authors as to whether stating this explicitly would be useful or not.

L324 - was anisotropy in the hyper-parameters considered? My prior assumption would be that the vertical and horizontal length scales could be quite different, but perhaps that was found not to be the case here?

Equation 15 - this is a really minor point, but just to make sure I've understood things - is $y$ not already included in the set $X$?

Figure 5 - I agree with RC1 - this would be best split into two separate figures. Also, the grey outline on the circles in Fig. 5a needs to be removed, as you currently have to zoom in a lot in order to see the fill colours of each point.

L431 - I'm not sure if this is the best place for it, but I think it is worth mentioning somewhere that there are alternative ways to deal with this smoothing problem. One approach is to select a variogram model that results in nearby points being assigned large weights (e.g. a linear model). Such a model must obviously be supported by the experimental variogram, but in any case a subjective choice must always be made regarding how the model parameters should be optimised to "best fit" the data. Therefore it can reasonably be justified that the model variogram should be chosen with a particular focus on representing the experimental data at small separation distances (see Kitanidis, 1997, for further discussion). A moving neighbourhood approach can also be adopted; in fact this is the default approach in the frequently used EasyKrig MATLAB package (e.g. O'Shea et al., 2014; Pitt et al., 2019). The cluster-based approach presented here has some advantages over these alternatives; in particular it removes many of the more arbitrary subjective choices associated with them. I think they are worth mentioning in this context, probably just a sentence or two would do.

L527 - Needs rephrasing. Could go for "As a general guideline, performing drone-based emission quantification of emission sources requires…"

L529 - Would it be clearer to say "at a downwind distance of less than 75 m"?

**References**

Alvarez, R. A., Zavala-Araiza, D., Lyon, D. R., Allen, D. T., Barkley, Z. R., Brandt, A. R., Davis, K. J., Herndon, S. C., Jacob, D. J., Karion, A., Kort, E. A., Lamb, B. K., Lauvaux, T., Maasakkers, J. D., Marchese, A. J., Omara, M., Pacala, S. W., Peischl, J., Robinson, A. L., Shepson, P. B., Sweeney, C., Townsend-Small, A., Wofsy, S. C. and Hamburg, S. P.: Assessment of methane emissions from the U.S. oil and gas supply chain, Science, 361, 186–188, doi:10.1126/science.aar7204, 2018.

Kitanidis, P. K.: Introduction to Geostatistics: Applications in Hydrogeology, Cambridge University Press, Cambridge, U.K., 1997.

O'Shea, S. J., Allen, G., Fleming, Z. L., Bauguitte, S. J.-B., Percival, C. J., Gallagher, M. W., Lee, J., Helfter, C. and Nemitz, E.: Area fluxes of carbon dioxide, methane, and carbon monoxide derived from airborne measurements around Greater London: A case study during summer 2012, Journal of Geophysical Research: Atmospheres, 119, 4940–4952, doi:10.1002/2013JD021269, 2014.

Pitt, J. R., Allen, G., Bauguitte, S. J.-B., Gallagher, M. W., Lee, J. D., Drysdale, W., Nelson, B., Manning, A. J. and Palmer, P. I.: Assessing London $CO_2$, $CH_4$ and CO emissions using aircraft measurements and dispersion modelling, Atmospheric Chemistry and Physics, 19, 8931–8945, doi:10.5194/acp-19-8931-2019, 2019.

---

## Author Comment (AC1)

**A tracer release experiment to investigate uncertainties in drone-based emission quantification for methane point sources**

Randulph Morales et al.

**Response to the Reviewer's Comments**

We thank the three reviewers for their positive comments, critical assessment, and useful points to improve the quality of our paper. In the following, we address their concerns point by point. Changes in the paper are shown in blue.

**Reviewer 1**

10

**5 General comments**

**Reviewer Point P1.1** — I recommend swapping th phrase 'tracer release' for something more appropriate such as 'controlled release', especially in the title. So-called tracer techniques rely on the concurrent measurement of the target gas (in this case methane), and a tracer gas, and are generally referred to as tracer ratio, tracer dispersion or tracer release methods in the literature (see e.g. Mønster et al., 2014; Yacovitch et al., 2017, and various references in the authors' own introduction). Such an experiment, involving a tracer gas, was not performed here. Validation of plume

mapping and flux methodology using a source with a known emission rate (as done in this work) is usually referred to as a controlled release experiment (see e.g. Thorpe et al., 2016; Heltzel et al., 2020; Shah et al., 2020).

**Reply**: We agree with the reviewer and changed all the existing 'tracer release' to "controlled release', including the title of the publication.

**15 A controlled release experiment to investigate uncertainties in UAV-based emission quantification for methane point sources**

**Reviewer Point P1.2** — Drone and UAV are used interchangeably throughout the text. It should be made more clear, somewhere early on, that the two refer to the same thing and one term (probably UAV) used consistently.

Reply: Changed as suggested

20 **Reviewer Point P1.3** — Figure text is often very difficult to read without considerably zooming in. All figure text needs to be much larger.

Reply: All figures are replotted and all font sizes are changed accordingly.

**Reviewer Point P1.4** — Calibration of the three instruments, with respect to measurements of methane, is not mentioned at all in the text. Was calibration to a methane standard (World Meteorological Organization) performed at all,

and if so what was the calibration procedure? If no calibration was performed, this throws the validity of the results into question.

**Reply**: This is an important point. We added descriptions of the calibration for the three instruments in their respective sections.

**1. Sect. 2.1: QCLAS**

- The instrument's precision, linearity, and calibration were described in detail elsewhere (Tuzson et al., 2020). Briefly, the instrument was calibrated by inserting it into a custom-built small volume (60 L) climate chamber. This chamber was then hermetically sealed and continuously purged with a certified calibration gas with high CH4 concentration (200 ppm ± 1%; PanGas, Switzerland). Furtheremore, the gas was dynamically diluted with dry nitrogen (N2) in a stepwise fashion using calibrated mass flow controllers.
  The overall uncertainty was estimated to be ±2%. Repeated experiments showed that the instrument preserves its linearity and only a marginal drift may appear in the offset. This, however, is fully accounted for, when applying the background CH4 subtraction step (see Sect. 4.4).
  - 2. Sect. 2.2: Active AirCore-CRDS:

A single-point calibration was used to correct the potential drift of the CRDS measurements. Measured methane mole fraction obtained using the AirCore system was linked to a known calibration standard that is traceable to the WMO X2004A  $CH_4$  scale (Vinkovic et al., in review )

3. Sect. 4.2:LI-COR-OTM-33A

The analyzer was calibrated before and after each measurement on the field and can be linked to at least two certified standards: the atmospheric  $CH_4$  value ( $2 \text{ ppm} \pm 5\%$ ), 5 ppm standard ( $5.05 \text{ ppm} \pm 5\%$ ), and a 25 ppm tank ( $24.98 \text{ ppm} \pm 5\%$ ).

45

40

**Reviewer Point P1.5** — The OTM-33A method usually uses instrumented vehicles and mobile sampling to quantify a flux, although examples do exist in the literature of stationary measurements (see e.g. Foster-Wittig et al., 2015; Shaw et al., 2020). Could the authors' comment on the possibility of applying the mobile OTM-33A method to the UAV-based measurements (has this possibility been explored)?

50 **Reply**: We have tested the OTM-33A approach on several drone flights during the ROMEO campaign (Röckmann and team, 2020). The method suggests the anemometer to be placed in approximately the same height and location as the

analyzer. As explained in P1.8, we tried to mount an anemometer (TriSonic Mini, Anemoment) on top of the drone but failed to isolate the wind data. Moreover, due to the limited flight time of the UAV, we could not fly the drone long enough to be able to obtain sufficient data necessary to provide an emission flux using the OTM-33A approach.

**Reviewer Point P1.6** — The plots appear to show CH4 mole fractions of 0 ppm (Figure 3 and 5). Unless these plots are 55 actually showing  $\Delta CH_4$  (CH4 – background), this is impossible. The tropospheric background mole fraction of methane is roughly 1.9 ppm (Lan et al., 2021). Could the authors explain these results?

**Reply**: All reported  $CH_4$  measurements in the manuscript are already above the background. We have changed the figure text into 'CH4 - CH4bg [ppm]' for easier comprehension. We added a paragraph discussing the background methane mole fractions as a response for P 1.16.

Reviewer Point P1.7 — Abbreviations for the six methods (CK, OK, PW, LW and combinations thereof) are used inconsistently. It would be useful to the reader for them to be introduced more distinctly in the methods section and then used consistently throughout the results and in figures/tables.

**Reply**: We added a few sentences in Sect. 5 to establish the six methods and their corresponding abbreviation.

65 ... A total of six guantification approaches were applied to all flights and evaluated for their ability to reproduce the true releases. These approaches arise from the combination of two different treatments of methane measurements and three different treatments of wind measurements. The treatments involved in mapping the discrete methane points into the measurement plane are the standard ordinary kriging (OK) and the cluster-based kriging (CK) interpolation schemes. The three different ways of estimating wind-speeds during 70 each guantification flight involves the scalar wind (SW), logarithmic wind (LW), and projected wind (PW) as discussed in Sect. 4.

75

60

**Reviewer Point P1.8** — Did the authors consider measuring wind speed and wind direction in-situ from the drone by attaching an anemometer? Concerning wind speed estimation/interpolation, would there be any improvement using a combination of methods two and three (LW and PW)? Further, in Section 4.3. I would recommend definitively stating the abbreviated definitions used to refer to the three approaches later in the text (for example "Proj. wind" Table 2, or "PW" later).

**Reply**: Prior to the controlled-release experiment, we tried to mount an anemometer (TriSonic Mini, Anemoment) on top of the drone to test whether wind measurements obtained from this set-up is a viable quantification approach. However, our tests suggested that wind measurements obtained this way are too noisy due to the interference from the UAV, and

we were not able to isolate the wind data. Regarding the combination of two different methods, the emission estimates 80 between the two methods are very close to each another, especially between the PW and LW, thus we don't expect a significant improvement by using a combination.

**Reviewer Point P 1.9** — How representative is the range of controlled release rates used here  $(0.2 - 0.7 \text{ g s}^{-1})$  of true emissions from oil and gas facilities, or other methane sources? I would expect real emissions to have a much greater range, and that the release rates used are at the lower end of that range. Are the authors' conclusions (for example, on wind speed, wind direction, and distance from plume limits) therefore only applicable to the controlled release rates used in this work, or are they equally applicable to emissions tens, or hundreds, of times stronger? If this is not the case, then the conclusions should be caveated by stating that these results are for a limited range of emission strengths.

Reply: A recent study by Omara et al. (2018) investigated over a thousand natural gas production sites in the US. In
their study, 85% of the sites belonged to a low- to mid-level natural gas production site category with emissions in the range of 0.13 – 0.58 g s-1 site-1. Although emissions per site are quite low, the sheer number of low-mid-level production sites accounted for almost two-thirds (63% [CI:45–83%]) of the CH4 budget in the US. "Super-emitters", producing an average of 2.31 g s-1 site-1, only accounted for 13% [CI:7-21%] of the total CH4 budget. Thus, we consider our results to be representative for the quantification of emissions from low-mid-level oil and gas wells. We have added this information
in Sect. 3: Control Release Experiment to put in context the chosen release rates during the experiment:

The release rates used in this study are a good representation of emissions from normal operating (i.e., excluding super-emitters) natural gas production sites in the US which produces  $0.13-0.58 \,\mathrm{g\,s^{-1}}$  (Omara et al., 2018).

**Specific comments**

105

100 **Reviewer Point P1.10** — L13: It is not clear to me what "stretched by 7 s and 0.06 seconds for every second of QCLAS measurement, respectively" is referring to here. This phenomenon is better explained in Section 5.3 and the authors should consider amending the abstract text to avoid confusion.

Reply: We have revised the abstract which now reads:

...smoothed by  $20 \,\mathrm{s}$  and had an average time lag of  $7 \,\mathrm{s}$ . AirCore measurements were also shifted linearly with time at an average rate of  $0.06 \,\mathrm{s}$  for every second of QCLAS measurement.

**Reviewer Point P1.11** — L105: For comparison, it would be useful to include the instrument measurement precision for the Picarro CRDS (as mentioned for the QCLAS system on L83).

**Reply**: We modified the text accordingly and added a sentence.

The precision  $(1\sigma, 0.25 \text{ Hz})$  of the CRDS was determined to be better than 0.7 ppb.

110 **Reviewer Point P1.12** — L129: Could "not too strong winds" be quantified here e.g. greater than X m s-1? I also assume this was due to the limitations of the UAV system used? Explaining the reason behind this limitation would be useful for guiding others.

Reply: We modified the sentence as follows:

..., i.e. days with no precipitation and a sufficiently large wind speed but smaller than  $8 \,\mathrm{m \, s^{-1}}$  which is the maximum value given by the UAV flight specifications.

**Reviewer Point P 1.13 — Figure 3**

- 1. The wind rose is exceptionally small and doesn't add much information to the figure in its current form. The wind rose might be better viewed in a separate panel, adjacent to the top-down view of the CH4 data. The wind rose is also not mentioned in the figure caption.
- Is the orange line showing the source-transect distance? This is not clear and should be made clear in the figure caption.
  - 3. The figure may benefit from an additional arrow illustrating the average wind direction for this flight (which ties in with the wind rose).

Reply: Figure 3 has been replotted.

- 125 1. The wind rose has been placed adjacent of the top-down  $CH_4$  data and caption was revised to include windrose
  - 2. Yes, the orange line is the source-transect distance. Now, it is explicitly mentioned in the caption.
  - 3. From the wind rose, one can infer the average wind speed and direction for this flight.

**Reviewer Point P 1.14** — L195: As for the Picarro instrument, it would be useful to include instrument characterisation (measurement precision etc.) here for the Li-COR instrument.

130 Reply: We have added some specifications of the instruments and added it on Sect. 4.2

The  $CH_4$  analyzer has a portable footprint  $(12 \text{ kg}, 51 \times 33 \times 18 \text{ cm}^3)$  and can measure methane mole fractions up to 50.0 ppm. It operates between -25 and 45 °C and can reach a precision  $(1\sigma)$  of 0.6 ppb at 1 s and 0.25 ppb at 5 s averaging time.

**Reviewer Point P1.15** — L219: Extra "to" in "matching the timestamp of the anemometer to the to GPS location".

**135 **Reply**: Correction applied.**

**Reviewer Point P 1.16** — L230: For clarification, were background CH4 mole fractions measured upwind of the emission source, or from either side of the emission plume? It might be useful to present the measured background mole fraction value(s) and uncertainty somewhere.

**Reply**: Background  $CH_4$  mole fractions were measured on either side of the emission plume. A discussion of measured background mole fraction was added and is now discussed in Sect. 4.4.

Background  $CH_4$  mole fractions were determined from measurements outside of the emission plume. Each sampled vertical height was extended to pass both sides of the plume to ensure sampling of local background values. Local variation of measured background values were corrected by using the Robust Extraction Base-line Signal (REBS) algorithm developed by Ruckstuhl et al. (2012). Average  $CH_4$  background mole fraction during the whole release experiment was determined at 2.09  $\pm$  0.19 ppm. Take-off and landing times of the UAV were noted and all data before and after the flight were removed.

Reviewer Point P1.17 — L237: Missing the word "to" between "due" and "the".

Reply: Correction applied.

Reviewer Point P1.18 — L271: Two identical references on this line - Tadić et al. (2015)

150 **Reply**: We corrected the identical reference.

Reviewer Point P1.19 — L285: Missing the word "one" between "only" and "cluster".

Reply: Change as suggested.

Reviewer Point P 1.20 — Figure 5

- 1. Could the caption include which instrument was used for the methane measurements shown?
- 155 2. The amount of panels here makes readability particularly difficult. I would recommend splitting into two separate figures: Fig. 5a as a single figure, and Fig. 5b, 5c, and 5d as a single figure. It may also be useful to have a direct side-by-side comparison of actual in situ measured CH4 (showing sparse spatial distribution on the vertical plane) alongside the 'predicted measured' Krigged CH4.
  - 3. Fig. 5b: As in general comments above, here CH4 mole fraction is in a range of -0.06 to +0.06 ppm. These values are impossible (especially the negative values) in the troposphere. Could the authors explain these results?

145

160

Reply: As suggested, we split Figure 5 into two separate figures.

- 1. Measurements were taken using the in-situ UAV-based QCLAS and this information was added in the caption of Figure 5.
- Figure 5 has been divided into two separate figures as suggested by the reviewer. A side-by-side comparison of measurements vs. predicted measured methane mole fraction is now shown in Fig. 6B and 6C.
  - 3. As mentioned, all CH4 reported in the manuscript are already above the background. The background values are discussed in Sect. 4.4

**Reviewer Point P1.21** — I would recommend moving equations 17 through 23 (and surrounding text) to a relevant section(s) in the methods section, as this is more Methodology than Results.

170 Reply: We moved subsection 5.1 into subsection 4.6: Method - Example of quantification procedure

**Reviewer Point P 1.22 — Table 2**

- This table is difficult to read due to the sheer amount of values. The information is much better visualised in a plot such as Figure 6. I would consider moving the full table to the Supplement, and only including the overall results (NMAE, Bias, RMSE) for all six methods in the main manuscript.
- 175 2. Abbreviations for the six methods (e.g. CKPW) are used throughout the text but not in this table.

**Reply:**

- 1. As suggested, we moved Table 2 in the supplement and only kept the overall results of the six quantification methods. We rewrote L387-388 to properly account for the changes in referencing the table:
- 180

165

The overall performance of each quantification approach is presented in Table 2 and estimated emission rates together with the true release rates for every individual flight are presented in Table S1.

2. The abbreviations for the six methods are included in the header of the table.

**Reviewer Point P1.23** — Figure 8: The caption should probably mention that these are residuals in flux estimates.

Reply: Changed as suggested.

**Reviewer Point P1.24** — Section 5.2.2: Could this section refer to Figure 8 as well, and the comparison of different meteorological regimes?

**Reply**: As suggested by Reviewer 2, we added a caption in figure 8 mentioning that optimal conditions and suboptimal conditions are defined in Section 5.1.2

**Reviewer Point P1.25** — Table 4: Would it be useful to present the NMAE, bias, and RMSE in this table, as done in comparisons of the AirCore with the QCLAS results (Table 3), and for the comparison of the six drone-based methods (Table 2)?

**Reply**: We have added the statistics in Table 4.

190

**Reviewer Point P1.26** — The link in the reference for US EPA 2014 goes to a page which states that "Emissions Measurement Center has Moved" – the link might need to be corrected.

Reply: The citation and the link have been updated.

195 **Reviewer Point P1.27** — Figure S6: Would it be worth showing the plume constructed from the QCLAS data too, for comparison?

**Reply**: The constructed methane plume for the QCLAS data is similar to the one shown in Fig. 6. Nevertheless, we added the constructed QCLAS methane plume in Fig. S6 to aid the comparison the difference between the two systems.

**References**

- 200 Foster-Wittig, T. A., Thoma, E. D., and Albertson, J. D.: Estimation of point source fugitive emission rates from a single sensor time series: A conditionally-sampled Gaussian plume reconstruction, Atmospheric Environment, 115, 101–109, https://doi.org/https://doi.org/10.1016/j.atmosenv.2015.05.042, 2015.
- Heltzel, R. S., Zaki, M. T., Gebreslase, A. K., Abdul-Aziz, O. I., and Johnson, D. R.: Continuous OTM 33A Analysis of Controlled Releases of Methane with Various Time Periods, Data Rates and Wind Filters, Environments, 7, https://doi.org/10.3390/environments7090065, https://www.mdpi.com/2076-3298/7/9/65, 2020.
- Lan, X., Nisbet, E. G., Dlugokencky, E. J., and Michel, S. E.: What do we know about the global methane budget? Results from four decades of atmospheric CH4 observations and the way forward, Philosophical Transactions of the Royal Society A: Mathematical, Physical and Engineering Sciences, 379, 20200 440, https://doi.org/10.1098/rsta.2020.0440, 2021.
- Mønster, J. G., Samuelsson, J., Kjeldsen, P., Rella, C. W., and Scheutz, C.: Quantifying methane emission from fugitive sources by
- 210 combining tracer release and downwind measurements A sensitivity analysis based on multiple field surveys, Waste Management, 34, 1416–1428, https://doi.org/https://doi.org/10.1016/j.wasman.2014.03.025, 2014.
  - Omara, M., Zimmerman, N., Sullivan, M. R., Li, X., Ellis, A., Cesa, R., Subramanian, R., Presto, A. A., and Robinson, A. L.: Methane Emissions from Natural Gas Production Sites in the United States: Data Synthesis and National Estimate, Environmental Science & Technology, 52, 12915–12925, https://doi.org/10.1021/acs.est.8b03535, pMID: 30256618, 2018.
- 215 Röckmann, T. and team, T. R.: ROMEO ROmanian Methane Emissions from Oil and Gas, EGU General Assembly 2020, https://doi.org/10.5194/egusphere-egu2020-18801, 2020.
  - Ruckstuhl, A. F., Henne, S., Reimann, S., Steinbacher, M., Vollmer, M. K., O'Doherty, S., Buchmann, B., and Hueglin, C.: Robust extraction of baseline signal of atmospheric trace species using local regression, Atmospheric Measurement Techniques, 5, 2613–2624, https://doi.org/10.5194/amt-5-2613-2012, 2012.
- 220 Shah, A., Pitt, J. R., Ricketts, H., Leen, J. B., Williams, P. I., Kabbabe, K., Gallagher, M. W., and Allen, G.: Testing the near-field Gaussian plume inversion flux quantification technique using unmanned aerial vehicle sampling, Atmospheric Measurement Techniques, 13, 1467–1484, https://doi.org/10.5194/amt-13-1467-2020, 2020.
  - Shaw, J. T., Allen, G., Pitt, J., Shah, A., Wilde, S., Stamford, L., Fan, Z., Ricketts, H., Williams, P. I., Bateson, P., Barker, P., Purvis, R., Lowry, D., Fisher, R., France, J., Coleman, M., Lewis, A. C., Risk, D. A., and Ward, R. S.: Methane
- flux from flowback operations at a shale gas site, Journal of the Air & Waste Management Association, 70, 1324–1339, https://doi.org/10.1080/10962247.2020.1811800, 2020.
  - Tadić, J. M., Ilić, V., and Biraud, S.: Examination of geostatistical and machine-learning techniques as interpolators in anisotropic atmospheric environments, Atmospheric Environment, 111, 28–38, https://doi.org/10.1016/j.atmosenv.2015.03.063, 2015.

Thorpe, A., Frankenberg, C., Aubrey, A., Roberts, D., Nottrott, A., Rahn, T., Sauer, J., Dubey, M., Costigan, K., Arata, C., Steffke, A.,

Hills, S., Haselwimmer, C., Charlesworth, D., Funk, C., Green, R., Lundeen, S., Boardman, J., Eastwood, M., Sarture, C., Nolte, S., Mccubbin, I., Thompson, D., and McFadden, J.: Mapping methane concentrations from a controlled release experiment using the next generation airborne visible/infrared imaging spectrometer (AVIRIS-NG), Remote Sensing of Environment, 179, 104–115, https://doi.org/10.1016/j.rse.2016.03.032, 2016.

Tuzson, B., Graf, M., Ravelid, J., Scheidegger, P., Kupferschmid, A., Looser, H., Morales, R. P., and Emmenegger, L.: A compact QCL

- 235 spectrometer for mobile, high-precision methane sensing aboard drones, Atmospheric Measurement Techniques, 13, 4715–4726, https://doi.org/10.5194/amt-13-4715-2020, 2020.
  - Yacovitch, T. I., Daube, C., Vaughn, T. L., Bell, C. S., Roscioli, J. R., Knighton, W. B., Nelson, D. D., Zimmerle, D., Pétron, G., and Herndon, S. C.: Natural gas facility methane emissions: measurements by tracer flux ratio in two US natural gas producing basins, Elementa: Science of the Anthropocene, 5, https://doi.org/10.1525/elementa.251, 69, 2017.

---

## Author Comment (AC2)

**A tracer release experiment to investigate uncertainties in drone-based emission quantification for methane point sources**

Randulph Morales et al.

**Response to the Reviewer's Comments**

We thank the three reviewers for their positive comments, critical assessment, and useful points to improve the quality of our paper. In the following, we address their concerns point by point. Changes in the paper are shown in blue.
* * *
**Reviewer 2**

**Response to the Reviewer's Comments**

**General comments**

**Reviewer Point P 2.1** — The Introduction/references, scope of work, and scientific approach of the work are good.

Some issues I found already mentioned by Reviewer 1 include finding L13 of the abstract confusing about the stretching by 0.06 seconds, wondering whether the methane data in all the plots are already background-subtracted, and suggesting improvements to the general readability of figures (other than Fig 1).

Regarding the novel cluster kriging approach adapted here, the paper cited by van Stein et al. (2020) concluded the method is designed to 'reduce the time and space complexity of the kriging method'. While dividing into elevated and background clusters makes sense, I do wonder how the above statement fits in. Specifically, if the difference between cluster kriging and ordinary kriging shown here has less to do with the theoretical basis of the method, and more a difference in the parameters used given that Fig 5d on left has a significantly different appearance (length scale or search radius?) than that of Fig 5c. Please explain. In general, while the math is presented if Section 4.5.2, I think some of the more practical details could be mentioned. Does the cluster kriging python package mentioned in the code availability statement also perform ordinary kriging, or that comes from elsewhere? Maybe add an example of the semivariogram or kriging parameters in the supplement to better illustrate the method?

**Reply**: We have revised the abstract to reduce the ambiguity in the text. All reported $CH_4$ mole fractions in the plots are enhancements above a background and we changed the labels of the plots to clarify this. Additional information on how background measurements were identified were added in Sect. 4.4. All plots were redone to improve readability and Fig. 5 was divided into two plots to reduce the amount of information in one single figure.

The average runtime to reconstruct the methane plume using the cluster kriging approach was around $150\,\mathrm{s}$ on a single core Intel Xeon W-2175 CPU @ 2.5 GH for 80x20 $\mathrm{m}^2$ grid at a $0.5\,\mathrm{m}$x$0.5\,\mathrm{m}$ resolution, while it took around $270\,\mathrm{s}$ to do the same using ordinary kriging—supporting the statetment of van Stein et al. (2020) on time-reduction. We agree, however, that the main motivation for applying the method was to account for the structural differences between the background field and the plume enhancements rather than computational aspects. The code also performs ordinary kriging. Regarding the significant difference between the cluster kriging vs ordinary kriging, the advantage of using the cluster kriging approach is the presence of the membership probabilities of the background and the elevated clusters. We have added Fig. S7 in the supplementary material illustrating the difference between cluster and ordinary kriging. The figure presents the reconstructed emission plume using ordinary kriging with length scales similar to that of the cluster kriging approach. It shows that the absence of membership probability of the background and elevated cluster in the ordinary kriging tends to produce noisier emission fields resulting in a 10% decrease in emission flux estimate.

**Specific comments**

**Reviewer Point P 2.2** — L27: Gurney et al. (2021) in Nature Communications is likely the wrong reference here. That paper is focused on FFCO2 from cities, not CH4 from oil and gas.

**Reply**: We have changed the reference and cited Alvarez et al. (2018); Omara et al. (2018); Zhang et al. (2020) for the text.

**Reviewer Point P 2.3** — L48-50: Shaw et al. (2021) and / or Hollenbeck et al. (2021) could be also considered adding here, as they are recent reviews on the subject of UAV methane quantification.

**Reply**: These are indeed relevant new publications, which are now cited in the manuscript.

**Reviewer Point P 2.4** — L114: 'capturing raw streams' – This is perhaps too vague. I think it is not so much a raw stream as a different stream (the carrier phase).

**Reply**: We edited the text to be more specific

> ... by capturing measurements of carrier phase signals from the GPS satellites ...

**Reviewer Point P 2.5** — Figure 2

1. The 14 on the x-axis tick labels is unneeded (see presumably matplotlib.dates.DateFormatter)

2. Isn't 0 AGL [m] defined as the takeoff altitude for UAV-GPS by the Matrice? Also, are takeoff and landing locations here different or the same?

3. The pressure altitude is impressively consistent comparing against RTK altitude. The spikes seen in the bottom panel of Figure (2) could be a little misleading since they appear to be caused simply by small differences in timing relative to the RTK during ascent and descent where altitude is changing quickly.

4. caption – the meaning of subscript m in bottom panel legend could be mentioned (slope from linear regression). They must also have some impact on the pressure altitude drift estimate unless robust regression was used

**Reply**:

1. Figure 2 has been replotted to remove the day "14". Labels have also been made bigger for better readability.

2. The 0 AGL $[\mathrm{m}]$ is defined as the takeoff altitude of the RTK-GPS. The takeoff and landing locations are not exactly at the same point but location of the two points is $\sim 2\,\mathrm{m}$ radius of each other.

3. To have a better representation of how the pressure altitude and UAV altitude differs from the RTK altitude, we have removed the data points related to the sudden ascent and descent of the UAV.

4. The caption text was adapted and we added a sentence at the end:

   Dashed blue and orange lines are fits representing a linear regression with the subscript *m* referring to the slope of the line.

**Reviewer Point P 2.6** — L144: Later, the make/model of 3D sonic anemometer is mentioned (uSonic-3 Scientific). What was the type of 2.5D anemometer?

**Reply**: The 2.5D anemometer is a TriSonic Mini from Anemoment. We have included the make and manufacturer of the 2.5D in the manuscript.

**Reviewer Point P 2.7** — Table 1 stability here is presumably based on equation from L215, not the Pasquill stability classes, which are also mentioned (L180)

**Reply**: Stability classes listed in Table 1 were indeed not determined according to Pasquill but by computing the turbulence parameters such as the Obukhov length and friction velocity. The Pasquill stability (PS) classes mentioned in L180 is a note to specify that the quantification procedure of OTM-33A was developed using the PS wind classes. We added this information in Sect. 3.

Stability classes listed in Table 1 were determined by calculating a dimensionless height, $\zeta = z/L$, where $z$ is the height of wind measurement and $L$ is the Obukhov length. The dimensionless height is used as a stability parameter where $\zeta < 0$ indicates unstable, $\zeta > 0$ unstable, and $\zeta = 0$ for neutral conditions.

**Reviewer Point P 2.8** — L250: It's a little unclear if the 3S algorithm is new to this manuscript, or if it is presented in the two manuscripts cited on L247 that are in preparation / review. With being able to read those, the writing here is a is a little hard to follow. A simple 1D Gaussian smoothing function need only have one parameter – a standard deviation. How does F(x,b) accommodate three parameters?

**Reply**: The 3S algorithm is not written in the two manuscripts that are in preparation and review. Thus, we have revised and added some clarifying text in the section.

We approximate the active AirCore measurement as, $\mathbf{y}$, defined as

$$\mathbf{y} = \mathbf{f}(\mathbf{x}, \mathbf{b}) + \mathbf{e} \tag{1}$$

where $\mathbf{f}$ is a model function that fits the high-resolution QCLAS and projects it onto the low-resolution AirCore measurement. The model function consists of $\mathbf{x}$ which is the independent variable where the QCLAS is measured (i.e., timescale) and the fit parameters $\mathbf{b}$ containing three elements describing the shift, stretch, and smoothing (i.e., 3S) of the AirCore. The error $\mathbf{e}$ represents the instrument's error as well as the error from the model function. We used a 1st-order Lagrange polynomial interpolation and applied a Gaussian filter to parametrize the shift, stretch, and smoothing of the AirCore. Starting with an arbitrary initial guess, the optimal parameters $\hat{\mathbf{b}}$ was determined using a nonlinear least squares fit solved iteratively using the Gauss-Newton method.

**Reviewer Point P 2.9** — Figure 4

1. Legend : 'Stretch' is written twice. Is one of them supposed to be shift? Also, the 0.06 s/s stretching is mentioned in abstract, but the other two numbers (12.81 and 17.90) are different?

2. Frankly, the algorithm mainly just seems to correct for the shift, also called time lag by some other authors. Are the other two parameters really helpful?

**Reply**: Figure 4 was redone to correct the legend

1. The numbers written in Fig. 4 are specific for that quantification flight. We derived the shift, stretch, and smoothing parameters for every individual flight where the QCLAS and the AirCore were present and took the average. The values written in the abstract are the average values.

2. We think that all three parameters are essential: The signal of the AirCore is smoothed out considerably due to mixing in the sampling tubes. Furthermore, the time of the AirCore measurement can only be indirectly determined, and our results showed that the time is not only be shifted by a constant value ("shift") but that the time lag may change over the duration of the flight ("stretch"). A reconstructed plume without using the stretch parameter for flight 312_03 is shown in the figure below.

[Figure]

**Figure 1.** Reconstructed methane plume for flight 312_03. The figure on the upper left shows a reconstructed plume from the QCLAS measurements using the cluster-based kriging approach. The figure on the upper right is the reconstructed plume without applying a proper time correction (i.e., no shift and no stretch) for AirCore measurements, whereas, the figure on the bottom left is a reconstructed plume obtained after applying the proper time correction (i.e., with both shift and stretch). The figure on the lower right is a reconstructed plume but only applying the time-lag and excluding the stretch (i.e., with shift and no stretch). By not accounting the stretch parameter, the methane plume split into two spatially.

**Reviewer Point P 2.10** — L361 16.04 kg should be g for the molar mass of methane

**Reply**: Correction applied

**Reviewer Point P 2.11** — Table 2: In footnote about optimal conditions, suggest mentioning they are defined in Section 5.2.2

**Reply**: We added this information in the caption, but we moved the full table to the supplement and only retained a summary of the results as suggested by Reviewer 1.

**Reviewer Point P 2.12** — Table 5: Suggest putting 'This study' (or similar) in the column next to Airborne CKPW mass-balance. Or somehow clarify, since the studies mentioned here - Golston et al. (2018); Yang et al. (2018); Shah et al. (2020) - do not use CKPW.

**Reply**: We have adopted the suggested change.

**Reviewer Point P 2.13** — L529: 'Under these conditions, measuring at a downwind distance of 75 m ensures the true emission can be fully mapped both horizontally and vertically'. This is a little confusing, since it sounds like you need to be $\geq$ 75 m downwind to fully capture the plume, while L523 indicates underestimation at those distances.

**Reply**: We have changed the text as suggested by reviewer 3.

...measuring at a downwind distance of less than $75\,\mathrm{m}$ ensures the true emission to be fully mapped...

**Reviewer Point P 2.14** — Figure 6, 8, and 9 show 'residuals' in %, which here must mean the percentage error of the estimate versus the known controlled release amount (but without calculating absolute values). Where does the 'range of residuals' come from?

**Reply**: "Range of residuals" shown in Fig. 6 was computed by taking the percentage error of the estimate–including the uncertainty of every individual flight–versus the known controlled release amount. In Figs. 8 and 9, the range of residuals only refers to the range of average residual (i.e., black dots in Fig. 6) computed for each quantification flight.

**Reviewer Point P 2.15** — L742 Suggest replacing the dead link to U.S. EPA with the new link

**Reply**: We have updated the U.S. EPA link

**References**

Alvarez, R. A., Zavala-Araiza, D., Lyon, D. R., Allen, D. T., Barkley, Z. R., Brandt, A. R., Davis, K. J., Herndon, S. C., Jacob, D. J., Karion, A., Kort, E. A., Lamb, B. K., Lauvaux, T., Maasakkers, J. D., Marchese, A. J., Omara, M., Pacala, S. W., Peischl, J., Robinson, A. L., Shepson, P. B., Sweeney, C., Townsend-Small, A., Wofsy, S. C., and Hamburg, S. P.: Assessment of methane emissions from the U.S. oil and gas supply chain, Science, 361, 186–188, https://doi.org/10.1126/science.aar7204, 2018.

Golston, L., Aubut, N., Frish, M., Yang, S., Talbot, R., Gretencord, C., McSpiritt, J., and Zondlo, M.: Natural Gas Fugitive Leak Detection Using an Unmanned Aerial Vehicle: Localization and Quantification of Emission Rate, Atmosphere, 9, 333, https://doi.org/10.3390/atmos9090333, 2018.

Gurney, K. R., Liang, J., Roest, G., Song, Y., Mueller, K., and Lauvaux, T.: Under-reporting of greenhouse gas emissions in U.S. cities, Nature Communications, 12, 1–7, https://doi.org/10.1038/s41467-020-20871-0, 2021.

Hollenbeck, D., Zulevic, D., and Chen, Y.: Advanced Leak Detection and Quantification of Methane Emissions Using sUAS, Drones, 5, https://doi.org/10.3390/drones5040117, 2021.

Omara, M., Zimmerman, N., Sullivan, M. R., Li, X., Ellis, A., Cesa, R., Subramanian, R., Presto, A. A., and Robinson, A. L.: Methane Emissions from Natural Gas Production Sites in the United States: Data Synthesis and National Estimate, Environmental Science & Technology, 52, 12 915–12 925, https://doi.org/10.1021/acs.est.8b03535, pMID: 30256618, 2018.

Shah, A., Pitt, J. R., Ricketts, H., Leen, J. B., Williams, P. I., Kabbabe, K., Gallagher, M. W., and Allen, G.: Testing the near-field Gaussian plume inversion flux quantification technique using unmanned aerial vehicle sampling, Atmospheric Measurement Techniques, 13, 1467–1484, https://doi.org/10.5194/amt-13-1467-2020, 2020.

Shaw, J. T., Shah, A., Yong, H., and Allen, G.: Methods for quantifying methane emissions using unmanned aerial vehicles: a review, Philosophical Transactions of the Royal Society A: Mathematical, Physical and Engineering Sciences, 379, 20200 450, https://doi.org/10.1098/rsta.2020.0450, 2021.

van Stein, B., Wang, H., Kowalczyk, W., Emmerich, M., and Bäck, T.: Cluster-based Kriging approximation algorithms for complexity reduction, Applied Intelligence, 50, 778–791, https://doi.org/10.1007/s10489-019-01549-7, 2020.

Yang, S., Talbot, R., Frish, M., Golston, L., Aubut, N., Zondlo, M., Gretencord, C., and McSpiritt, J.: Natural Gas Fugitive Leak Detection Using an Unmanned Aerial Vehicle: Measurement System Description and Mass Balance Approach, Atmosphere, 9, 383, https://doi.org/10.3390/atmos9100383, 2018.

Zhang, Y., Gautam, R., Pandey, S., Omara, M., Maasakkers, J. D., Sadavarte, P., Lyon, D., Nesser, H., Sulprizio, M. P., Varon, D. J., Zhang, R., Houweling, S., Zavala-Araiza, D., Alvarez, R. A., Lorente, A., Hamburg, S. P., Aben, I., and Jacob, D. J.: Quantifying methane emissions from the largest oil-producing basin in the United States from space, Science Advances, 6, eaaz5120, https://doi.org/10.1126/sciadv.aaz5120, https://www.science.org/doi/abs/10.1126/sciadv.aaz5120, 2020.

---

## Author Comment (AC3)

**A tracer release experiment to investigate uncertainties in drone-based emission quantification for methane point sources**

Randulph Morales et al.

**Response to the Reviewer's Comments**

We thank the three reviewers for their positive comments, critical assessment, and useful points to improve the quality of our paper. In the following, we address their concerns point by point. Changes in the paper are shown in blue.
* * *
**Reviewer 3**

**General comments**

**Reviewer Point P 3.1** — In general I've tried not to repeat comments already made by the other reviewers, but I do agree with Reviewer 1 that a brief discussion of calibration is required. I also think it would help to clarify things if the term "$CH_4$ enhancement" were to be used in cases where background values have been subtracted from the data (which I think is pretty much everywhere). Perhaps a couple of extra sentences briefly summarising the application of the REBS algorithm would be useful too (especially with regard to the Reviewer 1 question concerning where the background measurements were taken - I assume the answer is anywhere on the downwind measurement plane that the REBS algorithm identified)?

**Reply**: We agree that a brief discussion on the calibration of the instrument is needed to validate the measurements. We included a short text briefly discussing the calibration method of the instruments in their respective sections (Sect. 2.1, 2.2, and 4.2). The $CH_4$ molar fractions reported in the manuscript indeed refer to "$CH_4$ enhancement". To address this aspect, we adapted the text and referred to measured $CH_4$ as enhancements. We also changed all plot labels and used "$CH_4$ - $CH_{4bg}$ [ppm]" instead. Additional clarification was also added in Sect. 4.4 regarding how the background values were identified.

**Specific comments**

**Reviewer Point P 3.2** — L27: Alvarez et al. (2018) would be a more appropriate reference here (Gurney et al. (2021), is definitely wrong), although there are more recent options that would do the job too.

**Reply**: We replaced Gurney et al. (2021) by Alvarez et al. (2018) and also added the work of Omara et al. (2018); Zhang et al. (2020) in the citation.

**Reviewer Point P 3.3** — L202:the star on the friction velocity should be a subscript (as in Equation 4).

**Reply**: We have corrected the typo.

25 **Reviewer Point P 3.4** — L226:I'm a bit confused as to why this step was necessary. If both the QCLAS and RTK-GPS received GPS signals, why were they not already synchronised on GPS time?

**Reply**: This was just done to make sure that all the clocks across all systems, including the AirCore, have the same clock.

**Reviewer Point P 3.5** — L230:in addition to my general point above, it would probably be clearer to say that background
30 CH4 mole fractions were "subtracted" instead of "removed".

**Reply**: We have adapted the word "subtracted".

**Reviewer Point P 3.6** — Equation 5: I understand that this approach is based on previously published work, but the application is sufficiently different that it would be useful to provide some more information here. I suggest explicitly stating the form of F. Have I got it right that the parameter vector b consists of the QCLAS measurements? If so I would also
35 state that explicitly.

**Reply**: We have revised the text surrounding Eq. 5 to add further detail on the process of matching the AirCore data to the QCLAS data (see response to P2.8).

**Reviewer Point P 3.7** — Figure 4: Somewhere in either the caption and/or the associated main text it should be explicitly stated that these values were optimised separately for each flight.

40 **Reply**: We have fixed the text in Sect. 4.4.1: Processing of AirCore measurements

**Reviewer Point P 3.8** — L315 : maybe I missed something, but is it explained anywhere how the data are hard-clustered prior to performing ordinary kriging?

**Reply**: We have revised the text to explain this:

Hard clustered data-points are obtained by rounding the probability obtained from the GMM to either belong
45 to the background or the elevated cluster.

**Reviewer Point P 3.9** — L322 : I have no doubt that the Matèrn covariance kernel is a valid choice here, but as a general comment I feel that the choice of kernel should be based on an examination of the specific dataset on which kriging is being performed (although of course it can be guided by previous studies/experience). I'm sure that such examination was performed (i.e. someone checked to make sure the optimised function was a reasonable fit to the data for each flight) - I'm happy to leave it up to the authors as to whether stating this explicitly would be useful or not.

**Reply**: We did indeed try to use different covariance kernels (e.g., spherical, exponential, and gaussian among others) for our dataset and the best results were obtained with the use of a Matérn covariance kernel. The choice of using this kernel was reinforced when we came across the study of Stachniss et al. (2009) which also tested different covariance kernels in predicting a concentration field.

**Reviewer Point P 3.10** — L324 : was anisotropy in the hyper-parameters considered? My prior assumption would be that the vertical and horizontal length scales could be quite different, but perhaps that was found not to be the case here?

**Reply**: Anisotropy was not particularly considered in the optimization of the hyperparamaters.

**Reviewer Point P 3.11** — Equation 15 : this is a really minor point, but just to make sure I've understood things - is y not already included in the set X?

**Reply**: Yes.

**Reviewer Point P 3.12** — Figure 5 : I agree with Reviewer 1 - this would be best split into two separate figures. Also, the grey outline on the circles in Fig. 5a needs to be removed, as you currently have to zoom in a lot in order to see the fill colours of each point.

**Reply**: Figure 5 has been divided into two separate figures as suggested by reviewer 1. The grey outline on the circles are removed and the markers were also made bigger.

**Reviewer Point P 3.13** — L431 : I'm not sure if this is the best place for it, but I think it is worth mentioning somewhere that there are alternative ways to deal with this smoothing problem. One approach is to select a variogram model that results in nearby points being assigned large weights (e.g. a linear model). Such a model must obviously be supported by the experimental variogram, but in any case a subjective choice must always be made regarding how the model parameters should be optimised to "best fit" the data. Therefore it can reasonably be justified that the model variogram should be chosen with a particular focus on representing the experimental data at small separation distances (see Kitanidis, 1997), for further discussion). A moving neighbourhood approach can also be adopted; in fact this is the default approach in the frequently used EasyKrig MATLAB package (see e.g. O'Shea et al., 2014; Pitt et al., 2019). The cluster-based approach presented here has some advantages over these alternatives; in particular it removes many of the more arbitrary subjective choices associated with them. I think they are worth mentioning in this context, probably just a sentence or two would do.

**Reply**: Thank you for this information. We revised the text and moved it in Sect. 4.5.2: Kriging estimate

80

Although other kriging option modules are available such as a moving neighborhood approach where only data-points within a certain radius are considered in the kriging process (Mays et al., 2009; O'Shea et al., 2014; Pitt et al., 2019), the cluster-based kriging approach offers the advantage of removing many arbitrary subjective parameters present in other approaches.

**Reviewer Point P 3.14** — L527 : Needs rephrasing. Could go for "As a general guideline, performing drone-based emission quantification of emission sources requires. . . "

**Reply**: We have adapted the changes.

85 **Reviewer Point P 3.15** — L529 : Would it be clearer to say "at a downwind distance of less than 75 m"?

**Reply**: We have adapted the suggestion of the reviewer.

**References**

Alvarez, R. A., Zavala-Araiza, D., Lyon, D. R., Allen, D. T., Barkley, Z. R., Brandt, A. R., Davis, K. J., Herndon, S. C., Jacob, D. J., Karion, A., Kort, E. A., Lamb, B. K., Lauvaux, T., Maasakkers, J. D., Marchese, A. J., Omara, M., Pacala, S. W., Peischl, J., Robinson, A. L., Shepson, P. B., Sweeney, C., Townsend-Small, A., Wofsy, S. C., and Hamburg, S. P.: Assessment of methane emissions from the U.S. oil and gas supply chain, Science, 361, 186–188, https://doi.org/10.1126/science.aar7204, 2018.

Gurney, K. R., Liang, J., Roest, G., Song, Y., Mueller, K., and Lauvaux, T.: Under-reporting of greenhouse gas emissions in U.S. cities, Nature Communications, 12, 1–7, https://doi.org/10.1038/s41467-020-20871-0, 2021.

Kitanidis, P. K.: Introduction to Geostatistics: Applications in Hydrogeology, Cambridge University Press, https://doi.org/10.1017/CBO9780511626166, 1997.

Mays, K. L., Shepson, P. B., Stirm, B. H., Karion, A., Sweeney, C., and Gurney, K. R.: Aircraft-Based Measurements of the Carbon Footprint of Indianapolis, Environmental Science & Technology, 43, 7816–7823, https://doi.org/10.1021/es901326b, pMID: 19921899, 2009.

Omara, M., Zimmerman, N., Sullivan, M. R., Li, X., Ellis, A., Cesa, R., Subramanian, R., Presto, A. A., and Robinson, A. L.: Methane Emissions from Natural Gas Production Sites in the United States: Data Synthesis and National Estimate, Environmental Science & Technology, 52, 12 915–12 925, https://doi.org/10.1021/acs.est.8b03535, pMID: 30256618, 2018.

O'Shea, S., Allen, G., Fleming, Z., Bauguitte, S., Percival, C., Gallagher, M., Lee, J., Helfter, C., and Nemitz, E.: Area fluxes of carbon dioxide, methane, and carbon monoxide derived from airborne measurements around Greater London: A case study during summer 2012, Journal of Geophysical Research-Atmospheres, 119, 4940–4952, https://doi.org/10.1002/2013JD021269, 2014.

Pitt, J. R., Allen, G., J-B Bauguitte, S., Gallagher, M. W., Lee, J. D., Drysdale, W., Nelson, B., Manning, A. J., and Palmer, P. I.: Assessing London $CO_2$, $CH_4$ and CO emissions using aircraft measurements and dispersion modelling, Atmos. Chem. Phys, 19, 8931–8945, https://doi.org/10.5194/acp-19-8931-2019, 2019.

Stachniss, C., Plagemann, C., and Lilienthal, A. J.: Learning gas distribution models using sparse Gaussian process mixtures, Autonomous Robots, 26, 187–202, https://doi.org/10.1007/s10514-009-9111-5, 2009.

Zhang, Y., Gautam, R., Pandey, S., Omara, M., Maasakkers, J. D., Sadavarte, P., Lyon, D., Nesser, H., Sulprizio, M. P., Varon, D. J., Zhang, R., Houweling, S., Zavala-Araiza, D., Alvarez, R. A., Lorente, A., Hamburg, S. P., Aben, I., and Jacob, D. J.: Quantifying methane emissions from the largest oil-producing basin in the United States from space, Science Advances, 6, eaaz5120, https://doi.org/10.1126/sciadv.aaz5120, https://www.science.org/doi/abs/10.1126/sciadv.aaz5120, 2020.

---

## Referee Report (RR1)

The authors have done a good job in responding to the reviews – I suggest that this paper should now be published in AMT. I do have one remaining question, concerning the calibration of the QCLAS in the environmental chamber. I wonder if the use of dry nitrogen for this experiment is really the best option – does the different pressure broadening coefficient in dry nitrogen relative to wet air not bias the results? The field deployment data in Tuzson et al. suggests that any such effect does not result in a large bias, so I don't think the publication of this paper should be held up on this basis, but I would be interested to hear the authors' thoughts.